# SPX-related genes regulate phosphorus homeostasis in the marine phytoplankton, *Phaeodactylum tricornutum*

Kaidian Zhang[1,2], Zhi Zhou[3], Jiashun Li[1], Jingtian Wang[1], Liying Yu[1] & Senjie Lin [1,2,4 ✉]

Phosphorus (P) is an essential nutrient for marine phytoplankton. Maintaining intracellular P homeostasis against environmental P variability is critical for phytoplankton, but how they achieve this is poorly understood. Here we identify a SPX gene and investigate its role in *Phaeodactylum tricornutum*. *SPX* knockout led to significant increases in the expression of phosphate transporters, alkaline phosphatases (the P acquisition machinery) and phospholipid hydrolases (a mechanism to reduce P demand). These demonstrate that SPX is a negative regulator of both P uptake and P-stress responses. Furthermore, we show that SPX regulation of P uptake and metabolism involves a phosphate starvation response regulator (PHR) as an intermediate. Additionally, we find the SPX related genes exist and operate across the phytoplankton phylogenetic spectrum and in the global oceans, indicating its universal importance in marine phytoplankton. This study lays a foundation for better understanding phytoplankton adaptation to P variability in the future changing oceans.

[1] State Key Laboratory of Marine Environmental Science and College of Ocean and Earth Sciences, Xiamen University, Xiamen, Fujian, China. [2] Department of Marine Sciences, University of Connecticut, Groton, CT, USA. [3] State Key Laboratory of Marine Resource Utilization in South China Sea, Hainan University, Haikou, Hainan, China. [4] Laboratory of Marine Biology and Biotechnology, Qingdao National Laboratory of Marine Science and Technology, Qingdao, China. ✉email: senjie.lin@uconn.edu

Marine phytoplankton species contribute about 50% of the global primary production that provides organic carbon and oxygen to biota[1]. The productivity of phytoplankton relies on phosphorus (P) as one of the major nutrients[2]. However, in various parts of the world's oceans the primary form of P, dissolved inorganic phosphorus, is limited[3–5]. Phytoplankton species have evolved a set of strategies to cope with P limitation, including increase of inorganic phosphate (Pi) transporters, induction of hydrolases for scavenging organophosphates, and reducing P demand by replacing phospholipids with sulfur- or nitrogen-lipids[5–7]. A regulatory machinery must be in place to orchestrate these diverse pathways to achieve the synergistic outcome of P homeostasis in the phytoplankton cells, but our knowledge about the machinery is limited.

In plants, P homeostasis regulation involves SPX as an upstream negative regulator and PHR as an intermediate promoter of P acquisition mechanisms[8,9]. SPX is a functional domain (Pfam PF03105) named after yeast SYG1 (suppressor of yeast gpa1), yeast PHO81 (cyclin-dependent kinase inhibitor), and human XPR1 (xenotropic and polytropic retrovirus receptor 1)[10,11]. Depending on the absence or presence of additional domains, the family of SPX domain-containing proteins can be divided into four subfamilies: SPX, SPX-MFS, SPX-EXS, and SPX-RING[12]. SPX proteins refer to the proteins exclusively harboring the SPX domain[13]. Four SPX proteins in *Arabidopsis* and six in rice have been identified[8], nearly all of which (except AtSPX4 in *Arabidopsis* and OsSPX4 in rice) are responsive to Pi starvation[9]. SPX proteins act as negative regulators for P signaling and function in repressing the P starvation responses through a phosphate starvation response regulator (PHR)[14,15]. The PHR, a member of the Myb transcription factor (Myb TF) family, controls the expression of the majority of the phosphate starvation-induced (PSI) genes and influences numerous metabolic and developmental adaptations to P deficiency[16–18].

Within phytoplankton, the species in which P regulation has been best studied is arguably *Phaeodactylum tricornutum*, which has been widely used as a model species because its genome has been sequenced[19]. Recently, a Myb-like TF phosphorus starvation response gene (PSR, referred to here as PHR) was identified and shown by gene knockout experiments to act as a positive regulator of Pi signaling[20]. In addition, two SPX domain-containing proteins (named Vpt1 and Vtc4) have been identified in *P. tricornutum*[21]. Vpt1 (Phatr3_J19586), a vacuolar Pi transporter containing a SPX domain, is upregulated under P deficiency[22]. Vtc4 (Phatr3_J50019) is a SPX domain-containing vacuolar transport chaperone complex and its endomembrane localization is independent of P availability[21,22]. However, whether a regulatory cascade similar to plants' SPX-PHR[23] operates in *P. tricornutum* remains unclear. Here, we searched this species' genome and found that it possesses six SPX domain-containing genes, three of which (one SPX gene and two SPX domain-containing genes) were significantly induced by P deficiency. We focused on the SPX gene (Phatr3_J47434) and performed CRISPR/Cas9 mutagenesis and gene expression profiling coupled with physiological measurements to interrogate its functions in P regulation and functional association with PHR. We then mined a collection of reference transcriptomes derived from cultured marine eukaryotic microorganisms (MMETSP)[24], a catalog of genes derived from the Tara Oceans expedition (MATOU)[25], and a collection of metagenomics-based transcriptomes derived from the MATOU catalog (MGTs)[26] to examine the distribution and expression of the SPX genes across phytoplankton species and in the global oceans.

## Results

### SPX gene and CRISPR/Cas9 knockout.
Using the SPX domain as a query to search in the *P. tricornutum* genome, we found a total of six genes that harbor an SPX domain (Supplementary Table 1), including Vpt1 and Vtc4 that were recently described by Dell'Aquila et al.[21]. Pfam analysis of the six identified sequences in *P. tricornutum* resulted in the identification of the SPX domain in these proteins, which shared several conserved sites with land plants' SPX domains (Fig. 1a, b). Phylogenetic analyses further verified the high similarity of these sequences with other known SPX domain proteins (Supplementary Fig. 1). One of these genes possesses a SPX domain as the sole functional domain (named SPX, and its encoding gene named SPX gene, from here on) while the other five (including Vpt1 and Vtc4) contain at least one other domain. From our recently published transcriptome dataset[27], we found that SPX, Vpt1, and Vtc4 genes were differentially expressed ($\log_2$ Fold Change > 1 and adjusted $p$ value < 0.05). (Supplementary Table 1). To focus on determining the function of the SPX domain, only the SPX gene was chosen for mutagenesis, and the target site was located in the SPX domain but away from (upstream of) the conserved motif (Fig. 1c).

Five mutant strains of SPX (*m*SPX) were obtained. The mutation efficiency was 9.3%, and the mutation included various insertions and deletions (indels) at the target site (Fig. 1d–f). The deletion or insertion ranged from single nucleotide (nt), various short multi-nt, to a long tract (Fig. 1e). These indels are indicative of the non-homologous end joining (NHEJ) DNA repair mechanism operating in *P. tricornutum*. The four mutant clones shown in Fig. 1e had a deletion of 1 nt (clone 1-4 or *m*SPX1-4), 23 nt (clone 65-2), or 195 nt (clone 15-3 or *m*SPX15-3), or an insertion of 6 nt (clone 1-1), all but one causing a frameshift translation (*m*SPX1-4) or deletion of translation (*m*SPX15-3), and hence presumably functional loss or reduction.

The expression of SPX and its related genes in the mutants and WT grown under P+ were analyzed and compared using qPCR. When using primers located near the mutation site (F1/R1; Fig. 1c), qPCR results showed that SPX expression in *m*SPX1-4 and *m*SPX15-3 decreased by 4.4 folds and 6.2 folds, respectively (Fig. 1g). When using another pair of primers located in the 3′end region (F2/R2; Fig. 1c), far downstream of the mutation site, the expression of SPX in *m*SPX1-4 and *m*SPX15-3 exhibited no significant change compared to WT (Fig. 1g). The discrepancy between the two pairs of primers was most likely due to the influence of the mutation on F1/R1 priming efficiency in PCR. In addition, the other two P-stress inducible SPX domain-containing genes (Vpt1 and Vtc4) were significantly upregulated in both *m*SPX1-4 and *m*SPX15-3 compared to WT (Fig. 1g). To more systematically investigate the effects of SPX inactivation, a series of physiological and transcriptomic analyses were performed on the *m*SPX15-3 strain.

### Decreased growth rate, increased pigment contents and photosynthetic rate in SPX mutant.
Starting from the same initial cell concentration (400,000 cells ml$^{-1}$), the growth of *P. tricornutum* under different P conditions started to diverge one day later. P deficiency suppressed growth in both WT and *m*SPX15-3, whereas rapid growth occurred in both strains under P+ condition. Surprisingly, however, the *m*SPX15-3 grew slower than the WT under P+ condition (Fig. 2a). The average growth rate of *m*SPX15-3_P+ was $0.40 \pm 0.03$ day$^{-1}$ during the exponential growth phase (from day 1 to day 7), lower than that of the WT_P+ culture group, which was $0.47 \pm 0.02$ day$^{-1}$ (Fig. 2a).

Remarkable increases were observed in cellular pigment contents in the *m*SPX15-3 cells under both P + and P− conditions (Fig. 2b–d). After SPX mutation, chlorophyll *a* (Chl *a*) content increased by 14% (ANOVA, $F_{1,4} = 24.535$, $p = 0.008$) in P+ and 18% in P− culture groups (ANOVA, $F_{1,4} = 36.125$, $p = 0.004$), and carotenoid content rose by 22% (ANOVA, $F_{1,4} = 65.283$, $p = 0.001$) in P+ and 39% (ANOVA, $F_{1,4} = 50.548$, $p = 0.002$) in P− groups,

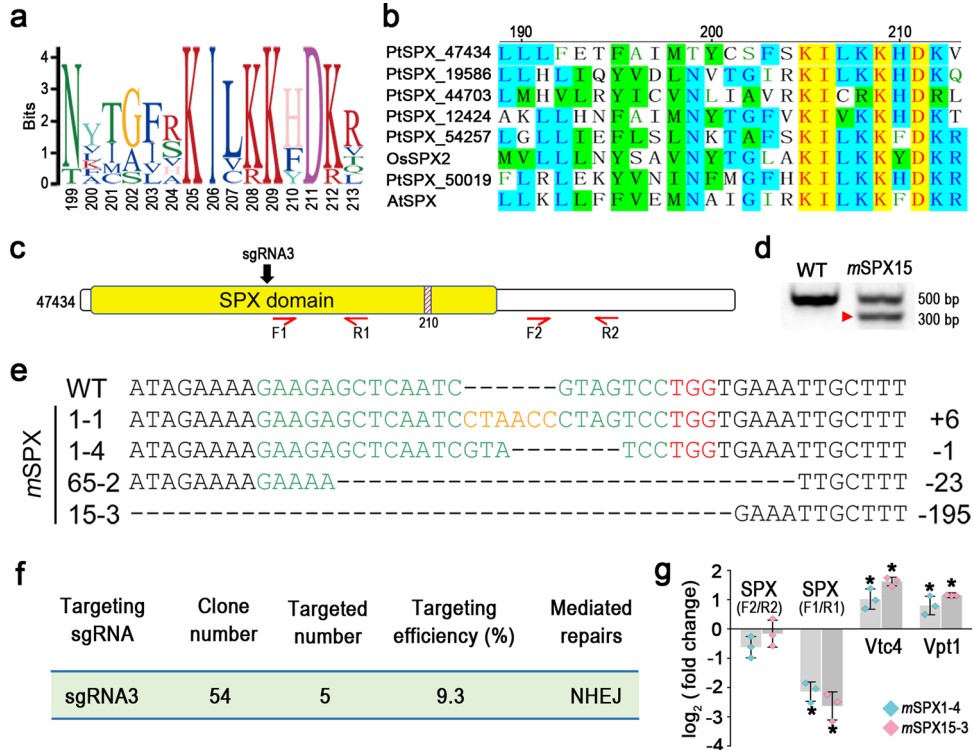

**Fig. 1 The identification and successful knockout of SPX gene. a** Graphical motif representation showing conserved amino acid residues of the SPX domain. **b** Alignment of SPX conserved sites with land plants. Identical residues are yellow-shaded while chemically similar ones are green-shaded. Included in the alignment are the six amino acid sequences from *P. tricornutum*, one from *Oryza sativa Japonica* Group (OsSPX2, XP_015614909.1), and one from *Arabidopsis thaliana* (AtSPX, NP_567674.1). **c** Schematic diagram showing the N-terminal SPX domain (yellow) and the effective sgRNA-targeting site (sgRNA3, arrow). The hatched area with number 210 indicates the location of the conserved domain shown in **a**, **b**. The red horizontal arrows indicate primers designed for RT-qPCR assay of SPX expression (results shown in **g**). F1 and R1 are located near the mutation target site whereas F2 and R2 are located far downstream from the mutation target site. **d** Target region of SPX gene amplified with specific primers, with the red arrowhead showing a deletion mutant band. **e** Sequence alignment of representative mutant clones with the wild type (WT). The sgRNA sequences, PAM sequences, and inserted sequences are labeled in green, red, and yellow, respectively. Clones of sequence 1-4 (*m*SPX1-4) and sequence 15-3 (*m*SPX15-3) were selected for downstream analyses. **f** Summary of on-target rate (targeting efficiency) in SPX knockout. NHEJ non-homologous end joining. **g** The expression of P stress-induced SPX-related genes in *P. tricornutum* *m*SPX1-4 and *m*SPX15-3 under P+ condition. Two SPX mutants (*m*SPX1-4 and *m*SPX15-3) were selected for comparisons with WT on the third day of inoculation into P+ growth medium ($n = 3$). Error bars, SD. Statistically significant changes in gene expression (*) in *m*SPX relative to WT is based on one-way ANOVA test ($p < 0.05$).

respectively (Fig. 2b, d). The elevation was also noted in chlorophyll *c* (Chl *c*), albeit without statistical significance (ANOVA, $F_{1,4} = 2.219$, $p = 0.211$ in P+; ANOVA, $F_{1,4} = 2.079$, $p = 0.223$ in P−) (Fig. 2c). Furthermore, photosynthetic rate, as determined as the rate of oxygen evolution, increased by 89% (ANOVA, $F_{1,4} = 58.638$, $p = 0.002$) in *m*SPX15-3 relative to WT under P− condition, although the effect was not detected under P+ condition (ANOVA, $F_{1,4} = 2.753$, $p = 0.172$) (Fig. 2e). Clearly, SPX mutation led to significant upregulation of photosynthetic capacity under P− condition.

**Increased cellular carbon content, nitrogen content, and neutral lipid content in SPX mutant**. Carbon content in *m*SPX15-3 cells increased by 22% (ANOVA, $F_{1,4} = 15.853$, $p = 0.016$) under P+ condition and 39% (ANOVA, $F_{1,4} = 146.728$, $p < 0.001$) under P− condition, respectively, relative to WT cells (Fig. 2f). Nitrogen content was higher by 24% (ANOVA, $F_{1,4} = 26.546$, $p = 0.007$) in *m*SPX15-3 cells than in WT cells under P+ condition, but no significant difference was observed under P− condition (ANOVA, $F_{1,4} = 1.212$, $p = 0.333$) (Fig. 2g). Consequently, the C: N ratio was not affected by SPX mutation under P+ condition (ANOVA, $F_{1,4} = 2.344$, $p = 0.200$) but increased (by 31%) under P− condition (ANOVA, $F_{1,4} = 63.385$, $p = 0.001$) (Fig. 2h).

The neutral lipid content per cell, as measured as fluorescence intensity of BODIPY 505/515 stain, increased by 5.83-fold (ANOVA, $F_{1,4} = 167.279$, $p < 0.001$) and 4.72-fold (ANOVA, $F_{1,4} = 2856.099$, $p < 0.001$) in WT and *m*SPX15-3, respectively comparing P− with P+ conditions (Fig. 2i), indicating that neutral lipid synthesis was markedly promoted by P deficiency. More notably, this lipid increase was magnified by SPX mutation, as the neutral lipid content in *m*SPX15-3 was 49% (ANOVA, $F_{1,4} = 47.547$, $p = 0.002$) and 21% (ANOVA, $F_{1,4} = 11.448$, $p = 0.028$) higher than in WT under P+ and P− conditions, respectively (Fig. 2i).

**Transcriptomic alteration caused by SPX mutation**. A total of 12 transcriptome libraries were constructed from four combinations of genotype and P conditions (*m*SPX15-3_P+, *m*SPX15-3_P−, WT_P+, and WT_P−, triplicated cultures for each) and sequenced to saturation. After removing low-quality sequences and adaptor sequences, on average 84.66% reads were uniquely mapped onto the reference genome, covering 93.24–94.44% of the genome-predicted proteome (Supplementary Table 2). Comparing *m*SPX15-3 with WT under both P conditions revealed 885 differentially expressed unigenes (Supplementary Data 1), indicating a broad impact of SPX mutation on the transcriptomic landscape. When *m*SPX15-3 was compared with WT separately for P+ and P− conditions, we found

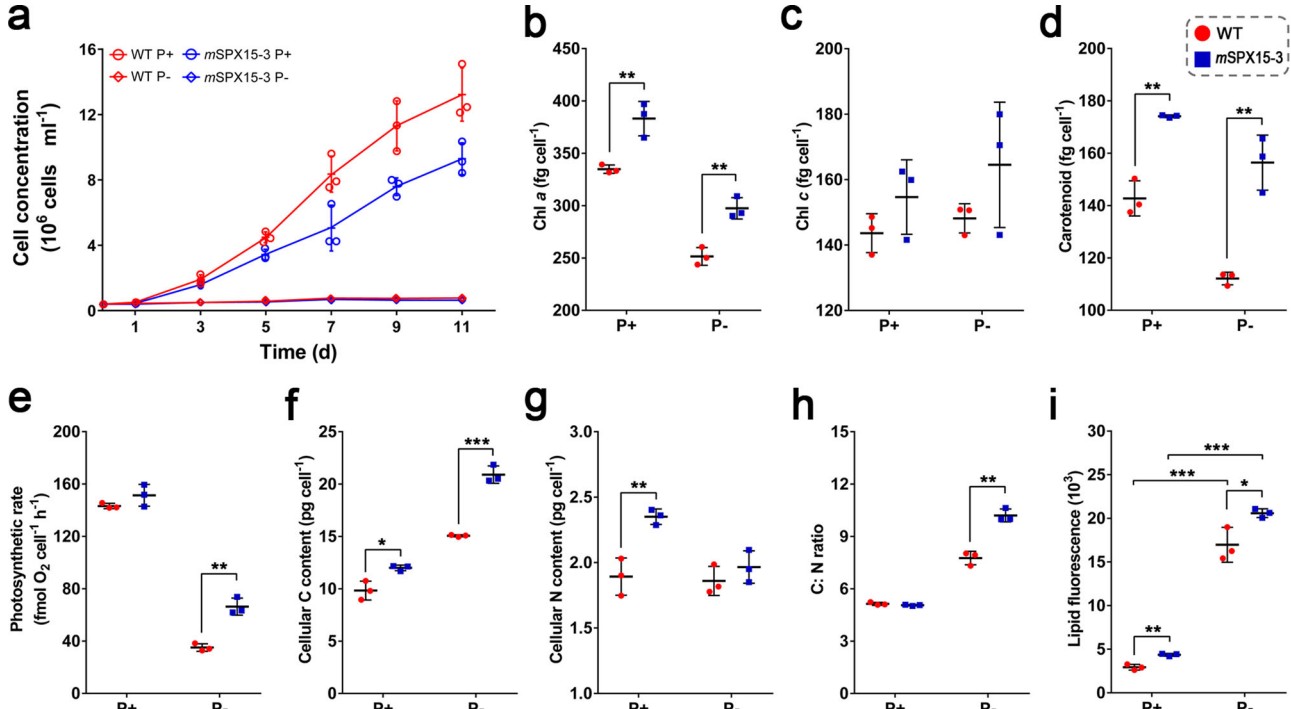

**Fig. 2 The physiological responses after SPX inactivation in *P. tricornutum* under different P conditions.** WT and *m*SPX15-3 strains were cultured in normal (P+) and P-deficient (P−) media. **a** growth curve; **b** chlorophyll *a* content; **c** chlorophyll *c* content; **d** carotenoid content; **e** photosynthetic rate; **f** cellular carbon content; **g** cellular nitrogen content; **h** cellular C: N ratio; **i** the intensity of neutral lipid BODIPY-stain fluorescence as measured by flow cytometry. Data are from day 3 of the (P+) or (P−) WT and *m*SPX15-3 cultures (*n* = 3). Error bars, SD. Asterisks indicate a significant difference, **p* value < 0.05, ***p* value < 0.01, and ****p* value < 0.001, ANOVA test.

528 differentially expressed genes (DEGs) for the P+ condition, 250 upregulated and 278 downregulated, and 560 DEGs for the P− condition, 306 upregulated and 254 downregulated (Supplementary Data 1). Of these, 203 DEGs appeared in both the P+ and P− conditions, indicating metabolic functions were influenced by SPX regardless of P nutrient condition. The 885 non-redundant DEGs represent metabolic processes regulated by SPX under P− or P+ conditions.

**Increased alkaline phosphatase (AP) activity and gene expression in SPX mutant.** *SPX* disruption led to significant increases in AP enzyme activity (Fig. 3a, b) and gene expression (Supplementary Table 3) under both P+ and P− conditions. The overall upregulated gene expression pattern was reflected in RNA-seq (Supplementary Table 3) and RT-qPCR, with similar magnitudes of elevation in the two different mutants examined (*m*SPX1-4 and *m*SPX15-3) (Fig. 3c). These results indicate that SPX is an upstream negative regulator of AP genes.

**Upregulation of Pi transporter genes in SPX mutant.** *SPX* mutation led to significant upregulation of P-responsive phosphate transporters (PTs) under P+ as well as P− conditions (Supplementary Table 4). Remarkably, the Nap$_i$3 (Phatr3_J47239) gene showed an 81.6-fold upregulation in *m*SPX15-3 under the P+ condition in our transcriptomic data (Supplementary Table 4), a trend also verified by RT-qPCR results in both mutants examined (Fig. 3c). In addition, one mitochondrial Pi transporter (PtMPT) was upregulated in *m*SPX15-3 under both P+ and P− conditions (Supplementary Table 4).

**Upregulation of phospholipid degradation genes in SPX mutant.** Eleven phospholipid degradation-related genes were found to be upregulated in *m*SPX15-3 compared to WT under either P+ or P−

conditions (Supplementary Table 5). Two glycerophosphoryl diester phosphodiesterases (GDPDs, Phatr3_J32057 and Phatr3_J49693) that participate in glycerophospholipid metabolism displayed strongly increased expression in both *m*SPX1-4 and *m*SPX15-3 mutants under P+ condition (Fig. 3c). In addition, one phosphoethanolamine phospholipase (Phatr3_J52110) showed upregulation in RNA-seq (Supplementary Table 5), which was not substantiated by RT-qPCR probably due to low expression levels overall (Fig. 3c). These results demonstrate that the SPX mutation causes the same effect as expected of P stress on WT, providing evidence that SPX is a negative regulator of phospholipid degradation, which is likely the first responder to P-nutrient dynamics.

**Relationship between SPX and PHR.** As a negative regulator for PT and AP as shown above, *SPX* is expected to be downregulated under P− condition. Surprisingly, *SPX* was upregulated instead in WT_ P− relative to WT_P+ (Supplementary Table 1). This suggests that AP and PT are not immediate targets of SPX; rather, an intermediate control mechanism may be at play between SPX and effectors AP and PT (Fig. 3d). We looked to Myb TFs because they are known to function as a central regulator of Pi starvation signaling in plants and algae[16]. We examined the nine Myb TFs that contain a single Myb domain (Myb1R) described by Rayko et al.[20,28]. Among these, we found that only Myb1R_5 (PHR) exhibited a significant response (upregulation) to both P-stress condition and SPX mutation (Supplementary Table 6), but not to PhoA or PhoD mutations[27,29], indicating that it functions downstream of SPX and upstream of APs. This is evidence that SPX-PHR is a coupled regulatory cascade of APs and PTs.

**Distribution and expression of SPX in all major phytoplankton phyla and across global oceans.** Based on a signature domain search on 306 pelagic and endosymbiotic marine eukaryotic species

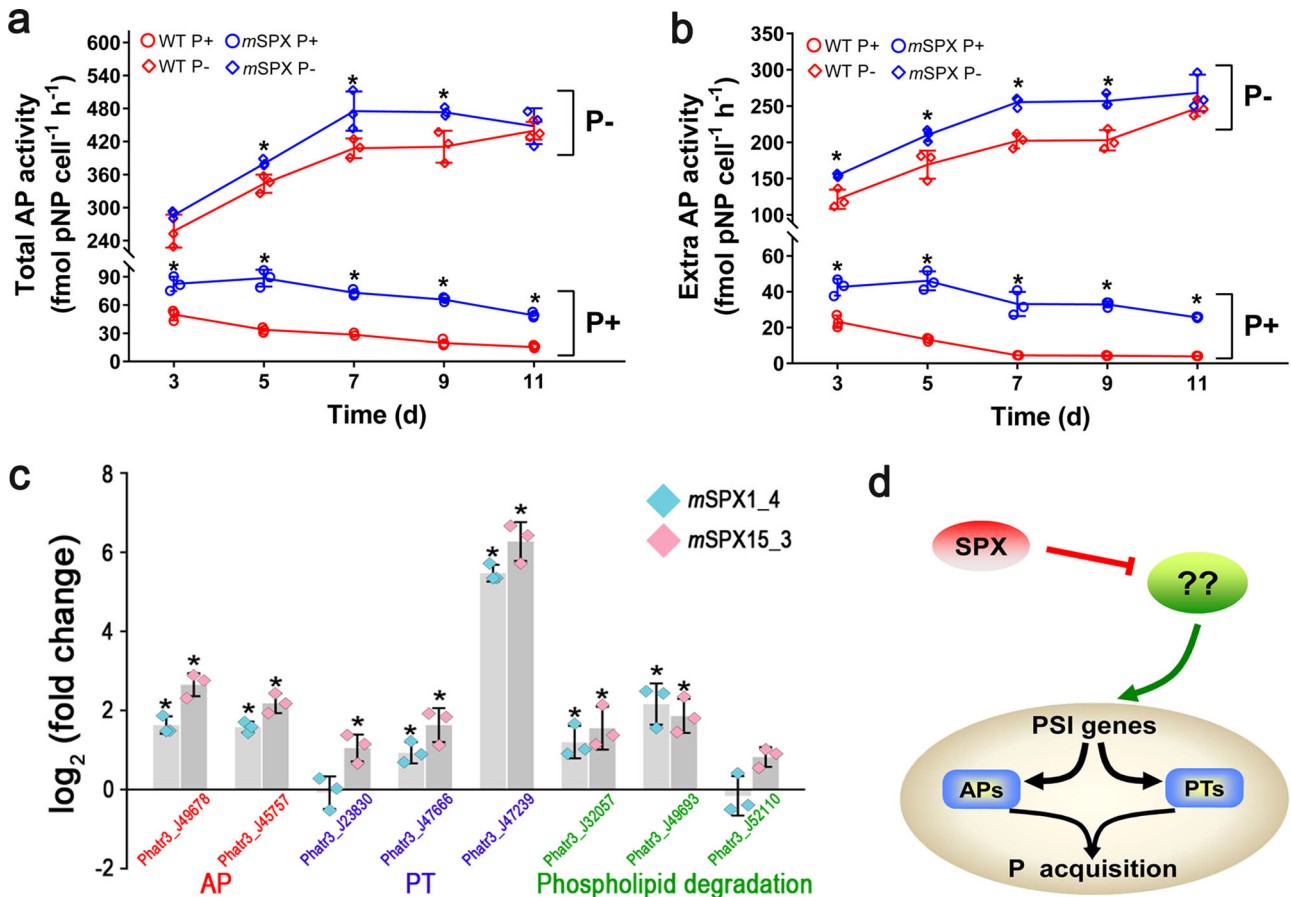

**Fig. 3 Evidence that SPX is an upstream negative regulator of AP and P transporters. a** Total AP activities in *m*SPX15-3 were higher than in WT.
**b** Extracellular AP activities in *m*SPX15-3 were higher than in WT. Error bars, SD. Asterisks in (**a**, **b**) depict significant differences ($n = 3$; $p < 0.05$).
**c** Changes in mRNA expression of AP, PT, and phospholipid degradation-related genes between SPX mutants (*m*SPX1-4 and *m*SPX15-3) and WT under P+
condition. Error bars, SD. Statistically significant changes (*) relates to comparison of *m*SPX with WT ($n = 3$; $p < 0.05$). **d** The putative model of SPX as an
upstream negative regulator in *P. tricornutum*. Positive and negative effects are indicated by arrows and flat-ended lines, respectively.

transcriptomes in the Marine Microbial Eukaryotic Transcriptome Sequencing Project (MMETSP) dataset[24], we found SPX domain-containing genes in 206 phytoplankton species that cover all major phytoplankton phyla (Supplementary Data 2). This indicates that the SPX gene is widespread in the phytoplankton tree of life, even though some of the detected genes might contain additional functional domains. Furthermore, we investigated the expression patterns and geographical distribution of SPX genes across Tara Oceans stations. The SPX gene was found to exist and be expressed widely across the global oceans, in size fractions from pico-, nano-, micro-, to small meso-plankton (Fig. 4a). Totally, 1131 SPX domain-containing genes were detected in the Marine Atlas of Tara Ocean Unigenes (MATOU-v1 catalog), of which 1091 belong to eukaryotes. They were identified at all of the 66 Tara Oceans stations (Fig. 4a) and were mainly (34%) distributed in the pico-eukaryote size fraction (0.8–5 μm), which were dominated by Chlorophyta in the subsurface layer (SRF) and Fungi in the deep chlorophyll maximum layer (DCM), respectively (Supplementary Fig. 2). In contrast, the other size fractions were dominated by Bacillariophyta (5–20 μm) and Metazoa (20–180 and 180–2000 μm) in both the SRF layer and DCM layer, respectively. Furthermore, the majority (96%) of the expressed SPX domains were concentrated in six lineages: Opisthokonta (Metazoa and Fungi), Stramenopiles, Haptophyceae, Viridiplantae (Chlorophyta), Rhizaria, and Alveolata, in decreasing order (Fig. 4b). In addition, 226 SPX domain-containing genes distributed in 58 species, were detected in 91 different Metagenomics-based Transcriptomes

(MGTs), two of which (MGT-15 and MGT-50) account for 34%. These expressed SPX domain-containing genes were most greatly contributed by the pelagophyte *Aureococcus anophagefferens*, the phaeophyte *Ectocarpus siliculosus*, and the chlorophyte *Chlamydomonas eustigma* (Supplementary Data 3). These results demonstrate the prevalence and hence importance of the SPX in global marine phytoplankton communities.

## Discussion

It is well recognized that phytoplankton species have evolved mechanisms to scavenge diverse sources of P to cope with P deficiency[30–32]. These mechanisms must function in concert to maintain P homeostasis in the cell while P availability in the environment fluctuates. This requires upstream regulators to function and orchestrate the genes involved in P uptake and metabolism. SPX domains in various proteins are well known to regulate P homeostasis and signal transduction in plants[9,13]. In this study, a SPX protein, which exclusively harbors the SPX domain, has been identified and functionally characterized in a phytoplankton. Our functional genetic manipulation, transcriptome profiling, and physiological observations in concert indicate the existence and operation of an upstream regulatory cascade in the model diatom *P. tricornutum*. Furthermore, by mining the TARA Oceans metatranscriptomic data, we found clear evidence that this SPX-based P-homeostasis regulatory mechanism likely exists in all major lineages of phytoplankton and is at play throughout the global oceans. The approach taken and the mutants generated here

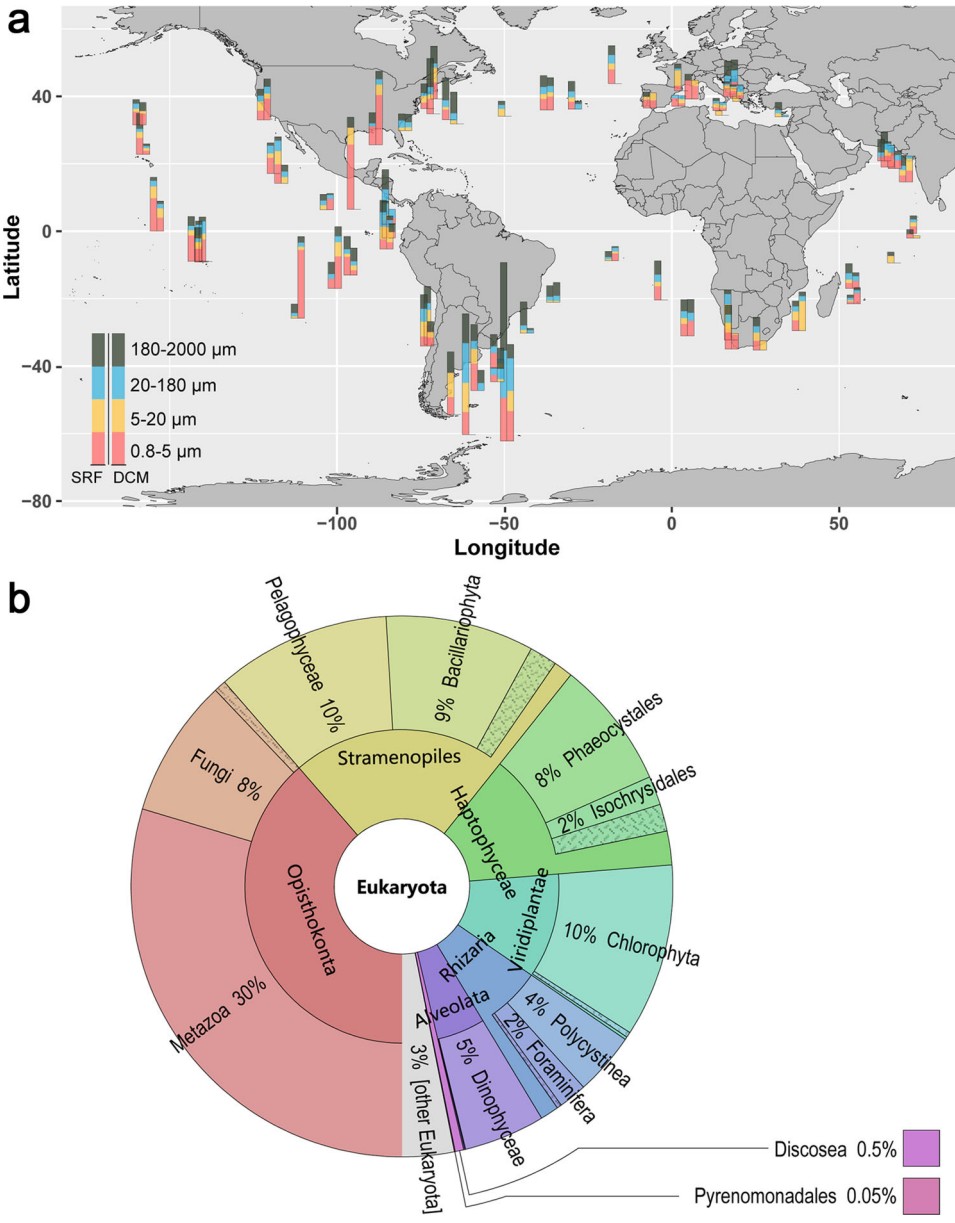

**Fig. 4 The distribution and expression of the SPX genes across phytoplankton species and in the global oceans. a** World map of the quantitative geographical distributions of SPX domain-containing genes. The expression values were computed as RPKM (reads per kilo base covered per million of mapped reads). The expression of SPX domain-containing genes were normalized to the total number of mapped reads. SRF subsurface, DCM deep chlorophyll maximum layer. The SRF layer and DCM layer were displayed on the left and right, respectively. Color of the squares depicts the size fractions of the sample. **b** Krona pie-charts showing the taxonomic distribution of SPX-domain-containing genes.

lay an important foundation for better understanding how phytoplankton species maintain P homeostasis in response to P variability and deficiency in various parts of the ocean.

Pi uptake is mediated by the abundance and efficiency of PTs, and usually most of the transporters are upregulated during P deficiency in microalgae[5]. Transcriptomic reconfiguration in mSPX15-3 was marked by the remarkable upregulation of Pi transporters under both P+ and P− conditions (Fig. 3c, Supplementary Table 4). This indicates that SPX has control over Pi transporter expression as a negative regulator. Therefore, SPX likely functions in the wild type *P. tricornutum* to constrain the capacity of phosphate uptake.

Another common response in phytoplankton to P deficiency is to increase the expression of phosphatases (most importantly AP) to

hydrolyze organophosphate and harness the released phosphate as a result[5]. In this study, AP activities and gene transcription were significantly elevated in mSPX15-3 (Fig. 3a–c, Supplementary Table 3). Besides, one acid phosphatase (ACP) (Phatr3_J44172) was co-induced in mSPX15-3 under P+ condition (Supplementary Data 1). ACP has been described as a constitutive enzyme, but in some algae has been shown to increase activities under phosphorus deficiency[33]. In addition, two genes encoding 5′-nucleotidase (Phatr3_J43694 and Phatr3_J44177), the enzyme usually upregulated in response to P deficiency in phytoplankton[34], were significantly upregulated in mSPX15-3 under both P+ and P− conditions (Supplementary Data 1). Nucleotides are a major source of intracellular P and the increased 5′-nucleotidases allow phytoplankton to scavenge P from DNA or RNA[35]. These results demonstrate that SPX places negative

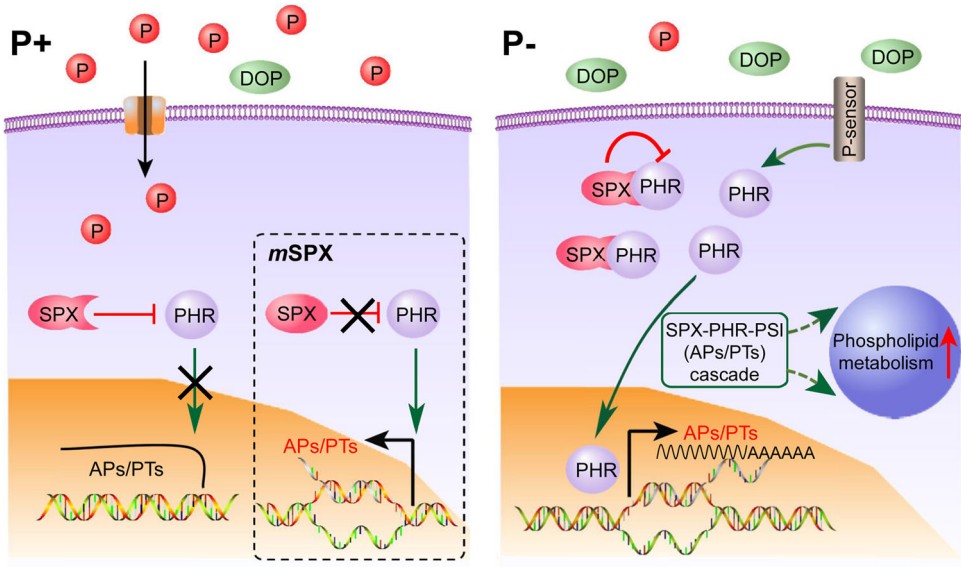

**Fig. 5 Schematic of action mode of the SPX-PHR-PSI regulatory cascade under P+ and P− conditions.** The P+ condition is shown on the left, in which regulation functions in the SPX mutant is shown in the inset, whereas the P− condition on the right. Positive and negative regulations are indicated by arrows and flat-ended lines, respectively. The vertical red arrow indicates upregulation. Crosses depict blockage of the pathway.

controls over organophosphate scavenging from the external environment (e.g. through extracellular AP) or through recycling inside the *P. tricornutum* cell.

Vacuoles serve as the primary intracellular compartment for Pi storage and remobilization in plants[36]. The transport of Pi across vacuolar membranes are crucial to buffering the cytoplasmic Pi homeostasis against external Pi fluctuations[37]. In plants, the SPX domain-containing major facilitator superfamily membrane transporter (SPX-MFS) proteins function as vacuolar phosphate transporters[38]. In this study, we found that a SPX-MFS protein, Vpt1, was significantly upregulated in *m*SPX15-3 under P+ condition compared with WT (Fig.1g, Supplementary Data 1). This suggests that SPX is an inhibitory regulator of Vpt1 in *P. tricornutum*. In addition, the vacuolar transporter chaperone (Vtc4), where the location of polyphosphate (polyP) synthesis occurs[39,40], was also significantly upregulated in *m*SPX15-3 under P+ condition compared with WT (Fig.1g, Supplementary Data 1). PolyP, formed from tetrahedral phosphate structural units, acts as an energy storage compound in diatom cells, and hence, when these cells sink, functions as a conveyor to pull down P for sequestration in the ocean floor[41]. Inferred from these, SPX protein potentially also function to maintain cytoplasmic Pi storage and homeostasis by regulating the other two SPX domain-containing genes in diatom *P. tricornutum*. If the same is true in other diatoms, this has the potential to influence the biogeochemical cycle of P in the ocean.

Upregulated phospholipid degradation is a common response in phytoplankton to P stress, along with increased synthesis of sulfur- and nitrogen-lipids, to decrease P demand[7]. In this study, *SPX* mutation resulted in increased expression of phospholipid degradation-related genes under both P+ and P− conditions (Fig. 3c, Supplementary Table 5). Therefore, in wild type *P. tricornutum*, SPX is also a negative regulator of phospholipid degradation. In marine phytoplankton, phospholipid degradation is usually accompanied by increased neutral lipid synthesis, primarily triacylglycerols (TAG), in response to P-stress[42]. Indeed, the neutral lipid content of *m*SPX15-3 cells significantly increased under both P+ and P− conditions (Fig. 2i). Accumulation of neutral lipids usually mirrors fatty acid synthesis in marine microalgae under nutrient deficiency[43]; however, contrary to the increased neutral lipids in *m*SPX15-3, 12 genes involved in fatty acid synthesis were significantly downregulated in

*m*SPX15-3 under P+ as well as P− conditions (Supplementary Table 7). These results strongly suggest that the increased neutral lipid content in *m*SPX15-3 was mainly due to the increased degradation of phospholipids producing more precursors for lipid accumulation rather than an increase in de novo synthesis of fatty acids. This is remarkable evidence that SPX in wild type *P. tricornutum* has intricate regulatory roles in lipid remodeling. This should be further verified in the future by analyzing lipid class changes in *m*SPX15-3.

In plants, SPX is an upstream inhibitory regulator, and PHR is a secondary and promoting regulator that relays SPX signals downstream to modulate PSI expression[8,14,15]. As recently reported, PHR in *P. tricornutum* controls conditional P acquisition and remodeling. Its knockout results in a significant decline of AP activity, PT expression, and phospholipid degradation[20], showing the opposite metabolic response relative to SPX mutation. Furthermore, the P stress-induced expression of PHR in WT and its upregulation in *m*SPX15-3 (Supplementary Table 6) places PHR as an intermediate between SPX and effectors (PSI) such as APs and PTs. It is plausible then that under P+ condition, SPX depresses PHR, thus maintaining low expression of AP and PT (Fig. 5). Under P stress, we find a greater increase in *PHR* than *SPX* expression, causing the PHR: SPX transcript ratio to increase from 1:9 under P+ to 1:4 under P− condition (Supplementary Table 8). As SPX presumably functions through binding to PHR[14], the relative increase of PHR may weaken the inhibitory signal of SPX, leading to elevated AP and PT expression, and increased phospholipid degradation (Fig. 5). This can explain the observed counterintuitive upregulation of SPX under P− condition. Further study is required, however, to verify this postulation or identify other regulatory elements that may exist to couple SPX and PHR in a more complex manner.

Strikingly, our data shows the SPX mutation also reshuffles nitrogen metabolism. The nitrogen metabolism in *P. tricornutum* is usually repressed under P− condition to echo a reduced demand for protein biosynthesis, which is often manifested by the downregulation of ribosome biogenesis[22]. In our study, SPX mutation caused upregulation of P acquisition machinery and a decrease in growth rate (Fig. 2a), resembling a P-stress response. However, the nitrogen content significantly increased in the SPX mutant grown under P+ (Fig. 2g). Furthermore, seven ribosome biogenesis related genes were also significantly upregulated in

*m*SPX15-3 (Supplementary Table 9), indicative of elevated nitrogen metabolism[22,44]. These changes are opposite to what would be expected of P-stress responses in WT cultures. Therefore, in addition to SPX-PHR-PSI cascade regulation, SPX in *P. tricornutum* potentially has a role in maintaining nitrogen metabolic homeostasis, which is worth investigating in the future.

The regulation of P homeostasis is crucial to photosynthetic organisms[45] as well as animals[46]. The widespread and active expression of the SPX gene in major lineages of phytoplankton and in the world's oceans (Fig. 4, Supplementary Data 2) allude to its important role in regulating P homeostasis in global ocean phytoplankton. We should note that because we used the conserved domain sequence as query, our search result likely not only include SPX genes but can also SPX domain-containing multi-domain genes. Future research should develop a more gene type specific algorithm to tease out how much each member of the SPX domain gene family contributes to this phylogenetically and geographically widespread P regulatory machinery.

## Methods

**Algal culture and experimental treatments**. *Phaeodactylum tricornutum* strain CCAP 1055/1 was obtained from the Culture Collection of Algae and Protozoa (Scottish Marine Institute, UK). The culture was maintained in autoclaved 0.22-μm filtered oceanic seawater, enriched with the full nutrient regime of the Guillard's F/2 medium[47], and made axenic by treatment with 1 X KAS compound antibiotics (50 μg ml$^{-1}$ kanamycin, 100 μg ml$^{-1}$ ampicillin and 50 μg ml$^{-1}$ streptomycin). Temperature was controlled at 20 °C and light was provided at a 14:10 light dark cycle with a photon flux of 120 μE m$^{-2}$ S$^{-1}$. All cultures in this study, including WT and mutants, were grown in the same way under these conditions except that in P treatments, where phosphate concentrations provided in the media varied.

For the P treatment experiments (for both the WT and the mutant, to be described below), a P-replete culture growing in logarithmic growth phase was inoculated into the low-phosphorus (< 5 μM) medium and allowed to grow until both external and intracellular P was depleted and growth stopped. This P− acclimated cells were then inoculated separately into P+ (36 μM phosphate) and P− (< 0.5 μM phosphate) media with an initial cell density of $4 \times 10^5$ ml$^{-1}$, each in triplicate. Cell counts and various parameters (to be described below) were measured daily from both the P+ and P− cultures.

**Identification of SPX domain-containing genes and phylogenetic analysis**. To look for SPX domain-containing genes in *P. tricornutum*, we used the conserved SPX domain sequence as query and carried out blast analysis against the existing *P. tricornutum* genome. Further Pfam domain searches using these six protein sequences were performed to examine whether the genes identified contained only SPX domain or additional domains were present. To verify the identity of the detected SPX genes, a phylogenetic tree was inferred based on the amino acid alignment of the SPX domain-containing genes from *P. tricornutum* and different organisms documented in NCBI.

BlastX was used for each gene copy against NCBI NR database and representative sequences within the top 30 hits (*e*-value of 3e−25) were selected for the phylogenetic analysis. The amino acid sequences were aligned by using MUSCLE[48] embedded in the software MEGA X[49], and edited manually using MEGA X. Prior to maximum likelihood (ML) phylogenetic analysis, the best amino acid substitution model for the SPX sequences was found to be LG + G[50]. The ML phylogenetic tree was constructed using MEGA X with 1000 bootstraps[49]. A Neighbor-Joining analysis conducted using MEGA X[49] yielded a similar tree topology.

**CRISPR/Cas9 knockout of SPX gene**. The PhytoCRISP-Ex, a CRISPR target finding tool, was used to design Cas9 target sites (G-N19-NGG) with low/no off-target potential[51]. Using the genome data for the analysis, off-target editing was prevented. The sgRNA sequence was designed (5′-GAA-GAGCTCAATCGTAGTCC-3′). The single guide RNA (sgRNA) adapter targeting SPX was ligated into pKS_diaCas9_sgRNA plasmid (Addgene ID: 74923) as described in Nymark et al.[52].

About $1 \times 10^8$ *P. tricornutum* cells were spread on 1.5% agar plates containing 50% seawater supplemented with F/2 medium nutrients. Transformation was performed using a Bio-Rad Biolistic PDS-1000/He Particle Delivery System (Bio-Rad, Hercules, California, USA) with a burst pressure of 1550 psi and a vacuum of 28 Hg[53]. The recombinant pKS_diaCas9_sgRNA plasmid (3 mg) was co-introduced with pAF6 plasmid (3 mg) carrying the zeocin resistance gene (Invitrogen, Thermo Fisher Scientific, Grand Island, New York, USA), with the former to induce mutation and the latter to facilitate selection of *P. tricornutum*[54]. Three mg of assembled pKS_diaCas9_sgRNA plasmid and 3 mg of the pAF6 selection plasmid were used for co-transformation. The bombarded cells were incubated on the agar plate under low light (50 μE m$^{-2}$ S$^{-1}$) at 20 °C. Two days later, cells were re-suspended in 600 μl sterile

50% seawater. About 200 μl of this suspension was plated onto agar medium containing 50 μg ml$^{-1}$ zeocin and 1 X KAS compound antibiotics, and the plate was incubated under a 14:10 light dark cycle at a photon flux of 120 μE m$^{-2}$ S$^{-1}$ at 20 °C. After three weeks, the resistant colonies were transferred to liquid medium containing 75 μg ml$^{-1}$ zeocin to 24-well plates (Bio-Rad), and cells were incubated for 10 days.

**Verification of SPX mutation using PCR and sequencing**. To screen for mutated cell lines, 100 μl of transformed cells were removed to PCR tubes. Tubes were centrifuged for 2 min and then re-suspended in 50 μl TE buffer (10 mM, pH 8.0). The re-suspended cells were frozen in liquid nitrogen, then heated immediately in 98 °C for 10 min using PCR Thermal Cycler (Bio-Rad). The freezing-heating cycle was repeated for 3 times. After a brief centrifugation, the supernatant for each clone was used directly or stocked at 4 °C. Three microliters of the cell lysates were used as template for PCR analysis using ExTaq DNA polymerase (Takara, Clontech, Japan). For checking whether the Cas9 component was successfully transformed into the cells, primer pairs were designed in the 3′ region of Cas9 sequence for use in PCR to amplify the Cas9 fragment (Supplementary Table 10). To confirm the disruption (insertions and deletions) of the target genes, the target regions were amplified using specific primers designed flanking the targets (Supplementary Table 10). Clean biallelic clones were initially identified by high-resolution melting (HRM) and verified by sequencing of the PCR products. The PCR products were first separated electrophoretically on a 1% agarose gel, purified using MiniBEST Agarose Gel DNA Extraction Kit (Takara), and cloned into pMD19-T plasmid (Takara) that was transformed into 5-alpha competent *E. coli* cells. At least ten random clones were picked for Sanger sequencing to get an overview of the different indels. Single nucleotide polymorphisms in the SPX gene were used to distinguish alleles in *P. tricornutum*[20,55]. After sequencing the target region, the cell lines with biallelic mutations were selected for further experiments.

**Comparison of expression levels of target genes between WT and SPX mutant**. Two *m*SPX clones (*m*SPX1-4 and *m*SPX15-3) and the WT were cultured under P+ condition, in triplicate. Around $2 \times 10^7$ cells from each culture were collected on the third day by centrifugation at 3000 *g*, 4 °C for 5 min, re-suspended in 1 ml Trizol Reagent (Thermo Fisher Scientific) for RNA work. RNA extraction, transcriptome sequencing, and differential gene expression analysis were conducted as described below to examine change of gene expression for SPX and other P-stress inducible genes between the WT and SPX mutants. To verify the transcriptomic data, RT-qPCR analyses were conducted using specific primers designed for these genes (Supplementary Table 10). For each sample, 200 ng of total RNA was reverse-transcribed in a 20 μl reaction mixture using the ImProm-II reverse transcriptase (Promega) with random hexamer primer according to the manufacturer's instructions. The cDNAs were diluted 1: 20 with nuclease-free water for further analysis. RT-qPCR was performed with a CFX96 Touch Real-Time PCR detection system (Bio-Rad) with iQ$^{TM}$ SYBR Green Supermix (Bio-Rad) following the manufacturer's recommendations. Due to the relatively constant expression of ribosomal protein small subunit 30S (RPS) under different conditions in *P. tricornutum*[56–58], RPS was used as the reference to calibrate the expression of related genes. Three technical replicate assays were performed for each RNA sample and their mean was used to represent their corresponding biological replicate.

**Measurement of cell concentration, growth rate, pigment contents, and photosynthetic rate**. The *m*SPX and WT cells were collected daily from each of the cultures to determine cell concentration. Cell counts were determined using a CytoFLEX flow cytometer (Beckman Coulter, USA) with 488/690 nm excitation light and emission light, respectively. The growth rate (μ) was calculated using μ = (ln $C_1$—ln $C_0$)/($t_1$—$t_0$), where $C_0$ and $C_1$ represent the cell concentrations at $t_0$ and $t_1$, respectively.

Pigment content for the *m*SPX15-3 and WT cells were determined spectrophotometrically. Cells from each culture were filtered onto GF/F membranes under gentle pressure and the membranes were immediately immersed into pure methanol and kept at 4 °C overnight in the darkness. After centrifugation at 5000 rpm for 10 min, the supernatants in each sample were used to measure the absorption spectra under a UV-VIS Spectrophotometer (Agilent Technologies, USA). The calculation of carotenoids, Chl *a*, and Chl *c* was performed according to the equations from Ritchie et al.[59].

The photosynthetic rate in *m*SPX15-3 and WT cells was determined by measuring the O$_2$ evolution rate at 20 °C with a Clark-type oxygen electrode (Hansatech, UK). The *m*SPX15-3 and WT cells were filtered onto 3 μm polycarbonate membranes (Millipore, USA) under low pressure. The collected cells were re-suspended into DIC-free artificial seawater (pH 8.05, buffered with 20 mM Tris-HCl) at a final concentration of ~3 × 10$^6$ cells ml$^{-1}$. About 1.5 ml cell suspension was dropped into the chamber with a photon flux of 120 μE m$^{-2}$ S$^{-1}$, which is the same as the daily light culture condition, to measure the net oxygen evolution rate. The rates of respiration were measured as O$_2$ consumption in the dark, and the photosynthetic rate was estimated as the sum of net oxygen evolution rate and respiration[60].

**Measurements of cellular carbon, nitrogen, and neutral lipid contents**. The *m*SPX15-3 and WT cells were filtered onto 25 mm GF/F membranes that had been burned in a Muffle Furnace at 450 °C for 5 h. Then the cell-containing membranes

were combusted in PE2400 SERIESII CHNS/O Elemental Analyzer (Perkin Elmer, USA), and the cellular C and N were measured. The measured C and N quantities in each sample were divided by the number of cells in the sample to yield C and N contents per cell, which were further averaged over the biological triplicate to obtain their respective means and standard deviations[61].

The lipid contents in $m$SPX15-3 and WT cells were quantified using flowcytometry. Cells were stained by adding 5 µl of 200 µg ml$^{-1}$ BODIPY 505/515 (Cayman Chemical, USA) DMSO solution into 1 ml culture of $1 \times 10^6$ cells ml$^{-1}$ cell density in the darkness at 25 °C, 20 min. BODIPY 505/515 is commonly used for semi-quantitative neutral lipids in microalgae[62,63]. To further compare changes in neutral lipid content after mutation, per-cell fluorescence was analyzed on a CytoFLEX flow cytometer (Beckman Coulter, USA) with 488 nm excitation and 510 nm emission.

**The measurement of AP activity**. AP activity of *P. tricornutum* was determined by adding 50 µl of 20 mM p-nitro-phenylphosphate (p-NPP from Fluka; dissolved in 1 M Tris buffer at pH 8.5) into 1 ml culture sample to yield final concentrations of 1 mM p-NPP and 50 mM Tris at pH 8.5[64]. Each reaction was carried out in a sterile 1.5 ml tube in the dark at 20 °C for 2 h. The samples were then centrifuged at 12,000 $g$ for 1 min. The supernatant was used for total AP activity measurement at 405 nm on a SpectraMax Paradigm plate reader (Molecular Devices, USA). For the measurement of extracellular AP activity, the culture was filtered through a 0.22-µm membrane and the filtrate was used to detect AP activity using the same spectrophotometric method as described above.

**Transcriptome sequencing**. As described above, about $2 \times 10^7$ cells were collected from each experimental culture, including the WT and SPX mutants under P+ and P− conditions (totally 12 cultures), in the exponential growth stage (on the third day after inoculation) using centrifugation at 3000 $g$, 4 °C for 5 min. One ml TRI-Reagent was added to each sample cell pellet. After vortex-mixing, the samples were immediately frozen and kept at −80 °C until RNA extraction was performed, always within a month. RNA extraction was carried out with Qiagen RNeasy Mini kit (Qiagen) and the RQ1 DNase (Promega) was used to eliminate the potential DNA contaminant. RNA concentration was quantified using NanoDrop ND-2000 Spectrophotometer (Thermo Scientific, Wilmington, DE, USA), and RNA integrity (RIN) was assessed using Agilent 2100 Bio analyzer. Samples with RIN > 7 passed the quality criteria and were sequenced on the BGISEQ-500 platform according to the manufacturer's instructions (BGI, Shenzhen, China). In total, 12 samples were sequenced, including WT under P+, WT under P−, mutant (clone $m$SPX15-3) under P+, mutant (clone $m$SPX15-3) under P−, each in triplicate, with one end read (50 bp).

**Identification of differentially expressed genes (DEGs)**. Raw reads for RNA-Seq were cleaned up by filtering low-quality reads (Q-score < 15 for > 20% of nucleotides in each read), trimming adaptor sequences and removing poly-N rich reads (> 5% of nucleotides in each read). After this, the remaining clean reads in the 12 samples were mapped to the *P. tricornutum* genome using HISAT software (http://ccb.jhu.edu/software.shtml)[65]. The genome sequences and annotation of *P. tricornutum* strain CCAP 1055/1 (http://ensemblgenomes.org/) were used as the reference. StringTie software (http://ccb.jhu.edu/software.shtml) and DESeq2 software package (http://www.bioconductor.org/packages/release/bioc/html/DESeq2.html) were used to estimate transcript abundances and identify genes differentially expressed between mutant and WT groups, respectively[65,66]. The Wald test in this study was used to calculate $p$ values, and then the $p$ values were adjusted for multiple testing using the procedure of Benjamini and Hochberg[67]. DESeq2 is based on the negative binomial distribution, and the parameters of DEGs were set to the absolute value of log$_2$ Fold Change > 1 and adjusted $p$ value < 0.05[66]. In total two different comparisons were analyzed in our study: DEGs in $m$SPX15-3_P+/WT_P+ comparison to explore the role of SPX in the P-replete condition, and DEGs in $m$SPX15-3_P−/WT_P− comparison to explore the SPX function in the P-deficient condition.

**Search for SPX expressed in other phytoplankton and in the global ocean**. To gain understanding on how important the SPX regulator is across the tree of phytoplankton phyla and in the global oceans, we examined their occurrence in MMETSP dataset, the TARA Oceans metatranscriptome dataset, and MGTs. The SPX domain (Pfam ID: PF03105) identified from the SPX protein was used as a query to search in the datasets. HMMER hmmsearch program was used for to search for the SPX gene using MMETSP dataset ($e$-value of 1e−10) and MGTs dataset ($e$-value of 1e−5), respectively.

For the TARA Oceans dataset from all major oceans except the Arctic during 2009–2013, hmmsearch was carried out using the MATOU-v1 catalog with a threshold of 1e$^{-5}$ to study their expression and biogeography[25]. The samples were collected from two representative depths: the SRF layer and the DCM layer, and planktonic eukaryotic communities were analyzed in four sizes: 0.8–5, 5–20, 20–180, and 180–2000 µm.

**Statistics and reproducibility**. In order to evaluate the statistical significance of the differences observed between WT and $m$SPX groups, one-way analysis of variance (ANOVA) was carried out using SPSS software (version 16.0, IBM, US). All values were given as Means ± SD with three biological replicates ($n = 3$), and statistical significance (*) was determined at the level of $p$ value < 0.05.

**Reporting summary**. Further information on research design is available in the Nature Research Reporting Summary linked to this article.

## Data availability
Raw sequencing data is available at NCBI in the SRA (Short Read Archive) database under accession number SRP214503. Source data corresponding to the figures are included in Supplementary Data 4, and uncropped gel image corresponding to Fig. 1d is included in Supplementary Fig. 3. All other data are available from the corresponding author upon reasonable request.

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

## Acknowledgements

We thank Ms. Chentao Guo for her logistic support. The work was financially supported by the Marine S & T Fund of Shandong Province for Pilot National Laboratory for Marine Science and Technology (Qingdao) (grant # 2018SDKJ0406-3) and Natural Science Foundation of China (grant #41776116). The Marine Microbial Initiative (MMI) of the Gordon and Betty Moore Foundation provided funding toward developing protist functional genetic tool (grant #GBMF grant #4980.01) that enabled S. L. to develop this work.

## Author contributions

S.L. conceived and supervised the work. K.Z. and S.L. designed the experiments and data analysis strategies. K.Z. and J.L. carried out the experiments. K.Z. and Z.Z. conducted data analyses. J.W. and L.Y. mined the MMETSP, TARA Oceans data, and MGTs. K.Z. and S.L. wrote the manuscript. All authors participated in revising the manuscript and agreed to the final submitted version.

## Competing interests

The authors declare no competing interests.
