## [Peer Review File · Communications Biology]

Reviewers' Comments:

Reviewer #1:

Remarks to the Author:

The manuscript "A regulatory cascade of phosphorus homeostasis in marine phytoplankton" by Zhang et al., describes a detection of SPX domain-containing genes in a model species of diatoms, *Phaeodactylum tricornutum*. The importance of SPX domain-containing proteins in Pi homeostasis and signaling has been demonstrated in a number of plants including *Arabidopsis*, soybeans, and rice but not in algae.

By using transcriptomics, CRISPR/Cas9 mutagenesis, and expression profiling coupled with measurements of enzymatic activity, the authors detected SPX domain-containing genes in *P. tricornutum* and assessed their functions in phosphorus regulation. They also mined publicly available metatranscriptomic data to examine the geographical distribution and expression of the SPX-PHR regulatory cascade in the global ocean.

I appreciate a combination of different approaches applied in this study including CRISPR/Cas9 mutagenesis, qPCR, and data mining techniques. However, I believe that certain sections of the manuscript would benefit from additional analyses that would strengthen the main conclusions of this study. In particular, I find that the data mining part of this work requires further attention and that additional computational analyses may further support the conclusions made by the authors. Please see my detailed comments below.

Comments:

Line 50. Please define SPX domain-containing proteins in the introduction section to help the reader understand why it was the focus of the manuscript.

Lines 53-54. MMETSP is a collection of transcriptomes representing organisms while MATOU is a gene catalog. Maybe it would make more sense to use the MGT collection (Metagenomics-based transcriptomes; please see my comment further below) in addition to the MATOU catalog to make the analysis more robust and logical.

Line 61. Briefly describe the methodology that you applied to find these genes from the metaT data.

Lines 107-109. I don't understand this sentence. Please check and rewrite.

Line 135. The Marine Microbial Eukaryotic Transcriptome Sequencing Project (MMETSP) contains 678 cultured samples of 306 pelagic and endosymbiotic marine eukaryotic species representing more than 40 phyla (Keeling et al. 2014).

Line 136. Did you have a look at the MGT catalog published earlier this year <https://genome.cshlp.org/content/early/2020/04/08/gr.253070.119>? The taxonomic diversity of the MGT collection differs significantly compared to the MMETSP project, hence you may strengthen your point stating the importance and wide distribution of the SPX and PHR genes.

Lines 140-142. I think more details would make this section stronger. All these data are publicly available and a more robust analysis should be carried out to support the conclusions of the importance of the discussed genes in the marine environments.

1. Which size fraction contributed the most to the overall expression of these genes? What taxonomic groups were found in this size fraction (SF)? How did taxonomic diversity vary across the size fractions, were the same organisms responsible for the expression of these genes or did different

organisms dominate the community depending on the SF?

2. Are you talking about expression of the genes (metatranscriptomics data) or their presence/abundance (metagenomics data)? I can see "abundance" in Fig 3a but it is described as "expression" in the text. Please clarify.

3. What do size fractions ">0.8 mkm" and "> 3mkm" mean? How are they related to others since they overlap?

Lines 143-144. They probably do but I think that a more thorough job needs to be done on the MMETSP/MATOU/MGT datasets to make and support this claim.

Line 152. Which physiological observations are meant here? And how do they point to the existence of this cascade? Maybe it is a terminology issue but I don't see any physiological results reported in the manuscript.

Lines 203-210. I find that this paragraph is somewhat repeating the previous one and it's a bit too vague overall. How exactly will this resource be valuable for future research? What exact questions will it help to answer?

Line 226. Usually a thorough description is provided on how transcriptomes are generated.

Line 234. Clearly making other knockouts and testing their physiology would strengthen the overall message of this paper. I understand that making knockouts may be laborious as it requires a lot of resources including time and funding but nevertheless, what are your plans regarding the SPX-like genes?

Line 332. Please define what is meant by low quality reads.

Line 528. From reference 37: "The abundance is estimated by the number of raw sequencing read nucleotides mapped to each gene from a gene catalog using MOCAT (Kultima, J.R. et al. 2012), normalised by one of the two methods selected in the submission form (normalisation method is recalled in the job details panel, see II.2 above for the description of the normalization schemes)."

Which normalization method was used here?

Lines 529-530. Why just DCM? Why not Surface (SRF) samples as well? Also, the fact that only DCM samples were chosen for the analysis should be explained in the Materials and Methods and should be mentioned in the results as well. However, I think that including both SRF and DCM samples would make more biological sense.

Fig 2c, 2e. What is presented on the Y axes, what is being normalized to RPS? And what is RPS?

Suppl Table 1. It says that q-values are "Equal to adjusted p-value, change is set at $q < 0.05$ in this study." However, adjusted p-values cannot be zero, they can only be approaching zero.

Reviewer #2:

Remarks to the Author:

The manuscript reports the characterization of an SPX gene in *Phaeodactylum tricornutum*, a model diatom for functional studies.

The authors first identify putative SPX genes from transcriptomics data, then choose one of them to produce CRISPR/Cas9 mutants and analyze the effects of SPX loss using enzymatic assays and gene expression analyses (RNA-seq and qPCR).

The main claim is that SPX is a negative regulator of phosphorous (P) uptake and of the response to P

starvation, working upstream of a phosphate starvation response gene (PHR). Wide conservation of these two genes in Tara Oceans data leads the authors to conclude that the mechanism proposed could be of universal importance for phytoplankton.

Genetic dissection and network reconstruction made via loss of function approaches have been difficult to accomplish in diatoms and have recently become approachable thanks to the CRISPR/Cas9 methodology. The experimental design in this study allows to propose a function for the SPX gene under study and to reconstruct a possible regulatory cascade, which appears to be similar to what is known for plants.

The findings therefore would add detail and depth to our understanding of nutrient metabolism in diatoms.

In many places however the manuscript lacks clarity and details should be provided to prove that the interpretation and conclusions are solid. A few relevant references are missing, and some of the important ones are cited but not really discussed.

I also suggest to perform a few more control experiments to support the findings.

Abstract

Line 22, "SPX protein" is not accurate, either "a member of the SPX family" or "an SPX gene"

Introduction

The introduction is too succinct, more information should be provided on the SPX gene family in general. Briefly, what data are available for plants? Where do SPX proteins localize? Is there a known protein architecture? What is the starting point for SPX identification in diatoms?

Also, I would explain why the SPX domain is called like that.

I think that the other KOs (phoA and phoD) that are used later in the paper should be introduced here, and more information on the data already available on genes involved in the P metabolism in *P. tricornutum* in general should be added. The gene that in this manuscript is named PHR is the same gene, named PtPSR, for which a KO has been published by Sharma et al., 2020, <https://nph.onlinelibrary.wiley.com/doi/full/10.1111/nph.16248>. This study should be introduced here, not simply cited, and the results obtained should be compared in the discussion.

References are completely missing in the text from line 47 to 57.

Results

Line 60 (and line 70, and 234), "SPX protein gene" does not make sense, perhaps "SPX genes"

Line 61 and 224-227, it is not clear if a name search or a homology-based search was made in the transcriptomics data to find SPX. Did the authors generate a de novo transcriptome from their data to look for the genes? If they did, where is it? Why did the authors use the transcriptomic data and not directly the genome to find the *P. tricornutum* genes?

Line 63, this is incorrect. Two of the genes in table S1 have been recently investigated in *Phaeodactylum*, see Dell'Aquila et al. 2020, doi: 10.3389/fpls.2020.00579.

Actually, there is a nomenclature proposed by Dell'Aquila et al. for part of the genes mentioned in the manuscript, would it be a good idea to use the same names? This would apply to the transporters too, and to PHR (Sharma et al., 2020).

Lines 64-65, the statement is not accurate as Fig.1a is showing the alignment of only one of the *P. tricornutum* SPX domain containing genes, all the other proteins shown in figure are from plants. It would be more informative to add all the *P. tricornutum* sequences and maybe remove some of the plant sequences, as the focus is on the newly identified *P. tricornutum* genes.

Line 67, supplementary figure 1, how were the sequences in the tree selected? There are plant sequences but also other diatom sequences, where do they come from? They must have been found in ncbi by blast, but no reference to this search can be found anywhere. Please add details.

The authors obtained five SPX mutant lines, which one did they use for the analyses and which mutations were present in the selected mutant? What was the effect of the mutations on the protein? Did they get a truncated protein, a protein with a missing part, or amino acid changes? It would be

informative to visualize the position of the conserved sequence shown in 1a in the scheme in 1b, to show if it is before or after the targeting site of sgRNA C.

Is there a predicted localization for the *P. tricornutum* protein? Can they check with specific software? The presence of a truncated protein or of a mutated protein could lead to different effects with respect to a complete loss of the protein (function partially retained, dominant negative effect, etc). There are Arabidopsis antibodies against SPXs, could the authors verify if they could work for this *P. tricornutum* SPX and possibly use them to check if a residual protein product is present? I think that the nature of the mutation in the protein could provide more information for the mechanism put forward in figure 4.

Is there a growth phenotype in the mutant?

The RNA-seq study and the qPCR analyses must be described better. Fig. 2c, d and e, at what time point were cell analyzed? Which phase of the growth curve was chosen? Were the qPCRs made on independent material or on the same RNAs used for the RNA-seq? Were other genes tested (and found not regulated) other than the ones shown in d and e? There is one gene in table S14 which does not appear in the paper.

Lines 90-97, a similar dataset was produced by Alipanah et al., 2018, <https://doi.org/10.1371/journal.pone.0193335>. I understand from the methods that the setup used here and in Alipanah were similar but not identical, anyway the purpose of finding genes involved in the P metabolism was the same, therefore I suggest to cite this study and comment on relevant differences (if any) in the discussion.

Line 97, change "rendering..." with "suggesting that the other five IPT could be..."

How are the high-affinity and low-affinity assignments made? Are there references to experimental data supporting the classification? I assume this is not based only on the gene expression profile. There is a tree in supplementary figure 2, some clades only contain diatom sequences (for instance the second cluster from the bottom), on which bases that clade is defined as high-affinity? Again in this tree there are sequences from other diatoms but it is not explained how they were selected and where they come from. In fact, how were the other organisms used in tree selected?

Back to the RNA-seq, I wonder why the authors focus on subsets of genes in the various supplementary tables, which is fine, but do not provide general data on the results of the gene expression analysis, especially when involving the mutant. How many genes were differentially expressed in total between the mutant and the wt? Are there other major functions disrupted? Do the samples cluster correctly in the routine quality controls of transcriptomics experiments? All these details should be given. I am also curious about whether any of the other SPX containing genes is regulated in the mutant.

Lines 124-125, how were the seven PHR genes identified?

To unequivocally demonstrate that the effects observed depend on the SPX mutation and not on background mutations in the mutant chosen for the RNA-seq, I would recommend to perform a few qPCRs on one of the other mutants to obtain further support to the specificity of the phenotype.

Line 140, the SPX and Myb domains and HMM profiles were used for the search, therefore what is found in the database are not only the orthologs to SPX (J47434) and PHR (J47256), but all the SPX and Myb containing genes, and some of these may have different functions. Without a more accurate analysis of the outputs obtained, it is not possible to claim that the two genes are expressed widely. The results simply indicate that the domains are broadly conserved.

Line 142, figure 3 is a screenshot of the output of the OGA blast, why are results shown for the deep chlorophyll maximum and not for the surface? In general, I think that screenshots are fine for a supplementary figure, but for a main figure I would try to provide something more informative. Can

the general result in the Tara oceans dataset be summarized differently?

Discussion

Since a KO is available for the other main partner in the model proposed (Sharma et al., 2020), the data should be cited and compared, and it should be discussed if they support the mechanism proposed in figure 4.

Line 169-170, the sentence is incorrect. Without measuring the lipids directly it cannot be claimed that phospholipid degradation is increased, there is only information on gene transcription in the paper.

Methods

Lines 231-232, the rationale for the choice of one of the six SPX containing genes should be placed earlier in the text and possibly better detailed, the reason for excluding the other two induced genes is relevant information.

Line 293, I do not understand why E. coli had to be replated.

Line 335, the entire description of how differentially expressed genes were obtained is missing, which method/software/parameters were used? Please add details.

Line 342, which method was used to analyze the qPCR results? Please add details.

Figure 2 and legend.

Why are the authors referring to "strains" in the legend? Is it a typo?

In this figure, if the qPCRs were made on the same material used for the RNA-seq (d) and (e) become less necessary (they confirm and repeat data found in tables) whereas it would be more interesting to see more of the general RNA-seq results. What is on the Y axis in d and e?

Minor details

Line 27, add an article before "phosphate"

Line 37, relies, not rely

Line 52, "its" referred to what? In the sentence before, SPX containing genes are mentioned.

Line 107, Seven

Line 167, remove the dash in up-stream

Lines 351 and 352, HMM and hmm, should it be the same?

Line 511, add an article before "deletion"

Line 514, specify what the green color is

Line 516, add an article before "upstream"

Line 528, SPX genes

Line 536, I do not see any text underlined or italicized in the figure

Line 537, regulation functions?

Line 540, arrow indicates

Reviewer #3:

Remarks to the Author:

In this paper, the authors investigate the role of SPX (and SPX-related) proteins in phosphate homeostasis in algae, using the diatom *Phaeodactylum tricornutum* as study case. According to what I have understood, the authors grew the alga in P-replete and P-limited conditions and then they produced transcriptomes in the same conditions. Such transcriptomes were used "to find" genes harbouring the SPX domains. Subsequently, they conducted a phylogenetic analysis to ascertain if such genes were homologous to the ones characterised in land plants. After positive results, they knocked out the SPX protein gene (Phatr3_J47434) and evaluated the effect of this alteration in the homeostasis of phosphate in mutant and wild type. Finally, using the metatranscriptomic data of Ocean Gene Atlas and the transcriptomes of MMETSP they searched for the presence of SPX genes across marine protists.

I think that this is an interesting paper contributing to the understanding of phosphate uptake and homeostasis in algae. The analyses are convincing of the results obtained but the way they are presented is confusing and difficult to follow. Indeed, after reading the manuscript several times, I still had difficulties in understanding some steps. For these reasons, I think that the manuscript needs some major revisions aiming at clarifying all the steps of the analyses and simplifying the reading before being considered for publication.

For example, I am not sure on what samples the transcriptomes have been sequenced. From the read, it seems that the authors have produced transcriptomes from P-replete and P-limited grown algae to analyse the differential expression of genes in these conditions. However, I could not find any reference to this in the manuscript (or not in a clear way). Then, at lines 90 on, it seemed to me that the transcriptomes were performed on the wild types and mutant algae. I think this need to be stated clearly in the manuscript.

Related to this point, I wonder if the authors have also found other genes (known or not from literature) differentially expressed in P-replete and P-limited conditions worthy to be at least discussed (and not used for mutagenesis).

Another major concern I have is into the Introduction. It is too short and the status of knowledge of P uptake and homeostasis in photosynthetic organisms (land plants, protists, etc) is barely introduced. The authors say at line 45 that "very little is known about the machinery in marine phytoplankton" but do not say anything about the machinery in other photosynthesising organisms. There is no reference to the role of SPX proteins (as well as other proteins and signalling molecules) in phosphate homeostasis and to recent literature about it. I only found a reference to Rubio et al. (2001). Authors should include recent literature and explain the current model, even if it is proposed for land plants. This could also be of help in the Discussion, since they conclude that a similar mechanism occurs in algae. Here some relevant examples:

Wild, R., Gerasimaite, R., Jung, J. Y., Truffault, V., Pavlovic, I., Schmidt, A., ... & Mayer, A. (2016). Control of eukaryotic phosphate homeostasis by inositol polyphosphate sensor domains. *Science*, 352(6288), 986-990.

Jung, J. Y., Ried, M. K., Hothorn, M., & Poirier, Y. (2018). Control of plant phosphate homeostasis by inositol pyrophosphates and the SPX domain. *Current opinion in biotechnology*, 49, 156-162.

Azevedo, C., & Saiardi, A. (2017). Eukaryotic phosphate homeostasis: the inositol pyrophosphate perspective. *Trends in biochemical sciences*, 42(3), 219-231.

Guo, J., Wilken, S., Jimenez, V., Choi, C. J., Ansong, C., Dannebaum, R., ... & Elrod, V. A. (2018). Specialized proteomic responses and an ancient photoprotection mechanism sustain marine green algal growth during phosphate limitation. *Nature microbiology*, 3(7), 781-790.

Related to the aforementioned literature, I wonder if the authors have found a differential expression in their transcriptomes of MYB transcription factors and light-harvesting related (LHCSR) proteins.

The Discussion is very short and not written in the context of recent literature. Therefore, I believe that the merits of the findings are not properly highlighted and I strongly recommend to rewrite this section as well as the introduction.

The materials and methods are extensively written but sometimes the workflow is not clear. I am not an expert of the Crispr/Cas approach but still found difficult to understand.

SPECIFIC COMMENTS

Line 22 = "we identify SPX protein". The sentence appears without any reference before to SPX proteins/domains and their role. Furthermore, later on the authors say that they found several SPX or SPX-like proteins.

Lines 49-50 = it is not clear to me if the authors were already searching for SPX genes in the transcriptomes or if they wanted to first ascertain if they were differentially expressed in the two different P-availability conditions and then to analyse their role in detail.

Line 61 = to what conditions the transcriptomes refer to?

Line 226 = here the authors state that they generated transcriptomes. Others? In what conditions? I think this is the point that confuses me.

Lines 283-285 = the concept is not clear. Please explain it better.

Line 334 = I do not understand what the authors mean for "quantify expression levels". From the mapping procedure? It sounds strange to me. Then after a few lines, the authors state that they have used RT-qPCR for quantifying the expression levels. Furthermore, for such analysis there is no specification of what the reference genes are and how the data normalisation has been performed.

Line 349. The authors have also searched for MYB domain and PHR proteins. Indeed, MYB transcription factor is known to be involved in phosphate starvation signalling in vascular plants and unicellular algae (see Rubio et al. 2001) but there is no reference to this in the Introduction.

Finally, the paper would benefit from some English check and editing, but I think that other major revisions are needed before reaching this point.

Dear Editors and Reviewers:

Thank you for your letter and for the reviewer's comments concerning our manuscript entitled "A regulatory cascade of phosphorus homeostasis in marine phytoplankton" (No.: COMMSBIO-20-2861-T). Those comments are all valuable and very helpful for improving our paper. With additional experiments conducted and new data added, we have addressed all the comments carefully and best to our capability. The manuscript was carefully revised according to the reviewer's suggestions. The main corrections in the paper and our point-by-point responses to the reviewer's comments are as follows:

In particular, please note that the following revisions would be necessary (at a minimum) for us to contact our referees again:

(1) Validate changes in mRNA and protein expression in SPX strains, as suggested by Reviewer #2. While we believe it would be interesting to generate additional lines, it would be more critical to comprehensively evaluate changes in expression or protein function among your existing strains.

Response: We have conducted new experiments on additional lines we have created, and measured changes of responses of genes regulated by SPX to SPX mutation, using qPCR (see page 5, lines 117-124).

(2) Incorporate the MGT transcriptome data set, as suggested by Reviewer #1.

Response: Done as suggested (see page 9, lines 240-244).

(3) Greatly elaborate on the transcriptomic analysis, overall methodology, and future implications or rationale for your research.

Response: Greater details have been provided about transcriptomic analysis (page 7, lines 165-176), overall methodology (page 19, lines 542-570), and future implications (page 10, lines 255-265).

(4) Carefully proofread the paper and improve the readability/presentation of results, as indicated by all three reviewers.

Response: The manuscript has now been carefully checked and corrected for English grammar.

Reviewer #1 (Remarks to the Author):

The manuscript "A regulatory cascade of phosphorus homeostasis in marine phytoplankton" by Zhang et al., describes a detection of SPX domain-containing genes in a model species of diatoms, Phaeodactylum tricornutum. The importance of SPX domain-containing proteins in Pi homeostasis and signaling has been demonstrated in a number of plants including Arabidopsis, soybeans, and rice but not in algae.

By using transcriptomics, CRISPR/Cas9 mutagenesis, and expression profiling coupled with measurements of enzymatic activity, the authors detected SPX domain-containing genes in P. tricornutum and assessed their functions in phosphorus regulation. They also mined publicly available metatranscriptomic data to examine the geographical distribution and expression of the SPX-PHR regulatory cascade in the global ocean.

I appreciate a combination of different approaches applied in this study including CRISPR/Cas9 mutagenesis, qPCR, and data mining techniques. However, I believe that certain sections of the manuscript would benefit from additional analyses that would strengthen the main conclusions of this study. In particular, I find that the data mining part of this work requires further attention and that additional computational analyses may further support the conclusions made by the authors. Please see my detailed comments below.

Comments:

(1) Line 50. Please define SPX domain-containing proteins in the introduction section to help the reader understand why it was the focus of the manuscript.

Response to reviewer comment No.1: We have now defined SPX domain-containing proteins as you suggested (page 3, lines 49-60).

(2) Lines 53-54. MMETSP is a collection of transcriptomes representing organisms while MATOU is a gene catalog. Maybe it would make more sense to use the MGT collection (Metagenomics-based transcriptomes; please see my comment further below) in addition to the MATOU catalog to make the analysis more robust and logical.

Response to reviewer comment No.2: The MGT collection analysis was added as you suggested (page 9, lines 240-244).

(3) Line 61. Briefly describe the methodology that you applied to find these genes from the metaT data.

Response to reviewer comment No.3: According to your suggestion, the detailed methodology was added to the Methods (page 14, line 400). Actually, the SPX domain-containing genes were identified in the *P. tricornutum* genome and transcriptome. First, from the genome data (Ensembl Protists) we found six genes harboring an SPX domain. Then, the P stress-induced expression of these SPX domain containing genes was examined from our transcriptomic data.

(4) Lines 107-109. I don't understand this sentence. Please check and rewrite.

Response to reviewer comment No.4: Revised as you suggested.

(5) Line 135. The Marine Microbial Eukaryotic Transcriptome Sequencing Project (MMETSP) contains 678 cultured samples of 306 pelagic and endosymbiotic marine eukaryotic species representing more than 40 phyla (Keeling et al. 2014).

Response to reviewer comment No.5: Modified according to your suggestion. Based on signature domain search, we found SPX related genes from 206 phytoplankton species among all 306 pelagic and endosymbiotic marine eukaryotic species in the MMETSP dataset.

(6) Line 136. Did you have a look at the MGT catalog published earlier this year <https://genome.cshlp.org/content/early/2020/04/08/gr.253070.119?> The taxonomic diversity of the MGT collection differs significantly compared to the MMETSP project, hence you may strengthen your point stating the importance and wide distribution of the SPX and PHR genes.

Response to reviewer comment No.6: Thank you for your suggestion. The MGT catalog analysis was added in the revised manuscript.

(7) Lines 140-142. I think more details would make this section stronger. All these data are publicly available and a more robust analysis should be carried out to support the conclusions of the importance of the discussed genes in the marine environments.

Which size fraction contributed the most to the overall expression of these genes? What taxonomic groups were found in this size fraction (SF)? How did taxonomic diversity vary across the size fractions, were the same organisms responsible for the expression of these genes or did different organisms dominate the community depending on the SF?

Are you talking about expression of the genes (metatranscriptomics data) or their presence/abundance (metagenomics data)? I can see "abundance" in Fig 3a but it is described as "expression" in the text. Please clarify.

What do size fractions ">0.8 mkm" and "> 3mkm" mean? How are they related to others since they overlap?

Response to reviewer comment No.7: Revised as you suggested. The SPX gene was found to exist and be expressed widely across the global oceans, in size fractions from pico-, nano-, micro-, to small meso-plankton. Totally, 1131

SPX-related unigenes were detected in the MATOU-v1 catalog, of which 1091 belong to eukaryotes. They were identified at all of the 66 Tara Oceans stations and were mainly distributed in the pico-eukaryote size fraction (0.8–5 μm). 96% of SPX expression was sequentially concentrated in six lineages: Opisthokonta, Stramenopiles, Haptophyceae, Viridiplantae, Rhizaria, and Alveolata.

In this study, we focus on the expression levels of SPX domain-containing genes. And the expression values were computed in RPKM (reads per kilo base covered per million of mapped reads). We have now checked and made sure we use clear wording.

In the revised version, the planktonic eukaryotic communities were analyzed in four sizes: 0.8-5 μm , 5-20 μm , 20-180 μm , and 180-2000 μm .

We have added all this information in the revised manuscript (page 9, lines 230-239; page 31, lines 872-879; page 20, lines 581-586)

(8) Lines 143-144. They probably do but I think that a more thorough job needs to be done on the MMETSP/MATOU/MGT datasets to make and support this claim.

Response to reviewer comment No.8: MGTs analysis was added as you suggested. And in the revised version, we focused on the distribution of SPX in all major phytoplankton phyla and across global oceans. Discussion of PHR is reduced to avoid distraction and confusion.

(9) Line 152. Which physiological observations are meant here? And how do they point to the existence of this cascade? Maybe it is a terminology issue but I don't see any physiological results reported in the manuscript.

Response to reviewer comment No.9: In the revised version, the cell concentration, growth rate, pigment contents, photosynthetic rate, cellular carbon and nitrogen contents, neutral lipid content, and alkaline phosphatase activity were added. The decreased growth rate, increased neutral lipid content, and increased AP activities in *mSPX* suggested the cells initiated the P-stressed

response after SPX inactivation under P+ condition. These results, together with transcription level analysis, indicate that SPX is an upstream inhibitory regulator in P regulatory network.

(10) Lines 203-210. I find that this paragraph is somewhat repeating the previous one and it's a bit too vague overall. How exactly will this resource be valuable for future research? What exact questions will it help to answer?

Response to reviewer comment No.10: Due to the extensive revision of the paper overall, and the section of Discussion in particular, this paragraph has been replaced by more elaborated information spread out in multiple subsections. We believe that significance of the work is more clearly articulated now.

(11) Line 226. Usually a thorough description is provided on how transcriptomes are generated.

Response to reviewer comment No.11: A detailed description of the transcriptome generation was added to the Methods.

(12) Line 234. Clearly making other knockouts and testing their physiology would strengthen the overall message of this paper. I understand that making knockouts may be laborious as it requires a lot of resources including time and funding but nevertheless, what are your plans regarding the SPX-like genes?

Response to reviewer comment No.12: Thanks for this comment. Exactly because the amount of time and resource to create a knockout, we chose to study the only SPX single-domain protein gene found in this species. The other two were SPX domain-containing genes contains other domains in addition (Vpt1 and Vtc4). Their functions will be studied in future projects by knockout experiments.

(13) Line 332. Please define what is meant by low quality reads.

Response to reviewer comment No.13: We have now provided information to

define low quality reads (page 20, lines 559-561).

(14) Line 528. From reference 37: "The abundance is estimated by the number of raw sequencing read nucleotides mapped to each gene from a gene catalog using MOCAT (Kultima, J.R. et al. 2012), normalised by one of the two methods selected in the submission form (normalisation method is recalled in the job details panel, see II.2 above for the description of the normalization schemes)."

Which normalization method was used here?

Response to reviewer comment No.14: In this study, the expression values were computed in RPKM (reads per kilo base covered per million of mapped reads). The expression of SPX domain-containing unigenes was normalized to the total number of mapped reads.

(15) Lines 529-530. Why just DCM? Why not Surface (SRF) samples as well? Also, the fact that only DCM samples were chosen for the analysis should be explained in the Materials and Methods and should be mentioned in the results as well. However, I think that including both SRF and DCM samples would make more biological sense.

Response to reviewer comment No.15: Both DCM and SRF samples were added in the revised manuscript as you suggested.

(16) Fig 2c, 2e. What is presented on the Y axes, what is being normalized to RPS? And what is RPS?

Response to reviewer comment No.16: RPS is small subunit ribosomal protein 30, the expression of which was measured and used as reference to which target gene expression was normalized. RPS was chosen as reference because the expression level of RPS is relatively constant under different conditions^{1, 2, 3}. In the revised version, we changed to absolute qPCR to relative qPCR, which is sufficient to monitor expression changes between *mSPX* and WT strains, and RPS

is still used as the internal reference gene.

(17) Suppl Table 1. It says that q-values are “Equal to adjusted p-value, change is set at $q < 0.05$ in this study.” However, adjusted p-values cannot be zero, they can only be approaching zero.

Response to reviewer comment No.17: Correct, and none of our values were actually zero, but in cases where it was very close to zero, we had to round it up to zero.

Reviewer #2 (Remarks to the Author):

The manuscript reports the characterization of an SPX gene in Phaeodactylum tricornutum, a model diatom for functional studies.

The authors first identify putative SPX genes from transcriptomics data, then choose one of them to produce CRISPR/Cas9 mutants and analyze the effects of SPX loss using enzymatic assays and gene expression analyses (RNA-seq and qPCR).

The main claim is that SPX is a negative regulator of phosphorous (P) uptake and of the response to P starvation, working upstream of a phosphate starvation response gene (PHR). Wide conservation of these two genes in Tara Oceans data leads the authors to conclude that the mechanism proposed could be of universal importance for phytoplankton.

Genetic dissection and network reconstruction made via loss of function approaches have been difficult to accomplish in diatoms and have recently become approachable thanks to the CRISPR/Cas9 methodology. The experimental design in this study allows to propose a function for the SPX gene under study and to reconstruct a possible regulatory cascade, which appears to be similar to what is known for plants.

The findings therefore would add detail and depth to our understanding of nutrient metabolism in diatoms.

In many places however the manuscript lacks clarity and details should be

provided to prove that the interpretation and conclusions are solid. A few relevant references are missing, and some of the important ones are cited but not really discussed.

I also suggest to perform a few more control experiments to support the findings.

Abstract

(1) Line 22, “SPX protein” is not accurate, either “a member of the SPX family” or “an SPX gene”

Response to reviewer comment No.1: We have changed the wording to “an SPX gene” when referring to the gene we knocked out.

Introduction

(2) The introduction is too succinct, more information should be provided on the SPX gene family in general. Briefly, what data are available for plants? Where do SPX proteins localize? Is there a known protein architecture? What is the starting point for SPX identification in diatoms?

Also, I would explain why the SPX domain is called like that.

I think that the the other KOs (phoA and phoD) that are used later in the paper should be introduced here, and more information on the data already available on genes involved in the P metabolism in P. tricornutum in general should be added. The gene that in this manuscript is named PHR is the same gene, named PtPSR, for which a KO has been published by Sharma et al., 2020, <https://nph.onlinelibrary.wiley.com/doi/full/10.1111/nph.16248>. This study should be introduced here, not simply cited, and the results obtained should be compared in the discussion.

Response to reviewer comment No.2: We have now introduced SPX and PHR (PtPSR) with details in Introduction. Specifically, the terminologies of SPX and its various multi-domain proteins are explained.

(3) References are completely missing in the text from line 47 to 57.

Response to reviewer comment No.3: We have now expanded the paragraph and provided references.

Results

(4) Line 60 (and line 70, and 234), “SPX protein gene” does not make sense, perhaps “SPX genes”

Response to reviewer comment No.4: Changed as suggested.

(5) Line 61 and 224-227, it is not clear if a name search or a homology-based search was made in the transcriptomics data to find SPX. Did the authors generate a de novo transcriptome from their data to look for the genes? If they did, where is it? Why did the authors use the transcriptomic data and not directly the genome to find the *P. tricornutum* genes?

Response to reviewer comment No.5: Sorry for the lack of clarity before. We have now made it clearer by briefly describing the finding of SPX domain-containing genes from the genome using BLAST with SPX domain sequence as query. Subsequently, we used the transcriptomic data we generated de novo under P+ and P- growth conditions to characterize expression dynamics of the SPX domain containing genes. See page 14, lines 400-414.

(6) Line 63, this is incorrect. Two of the genes in table S1 have been recently investigated in *Phaeodactylum*, see Dell’Aquila et al. 2020, doi: 10.3389/fpls.2020.00579.

Actually, there is a nomenclature proposed by Dell’Aquila et al. for part of the genes mentioned in the manuscript, would it be a good idea to use the same names? This would apply to the transporters too, and to PHR (Sharma et al., 2020).

Response to reviewer comment No.6: We have updated literature and use of terminology. In the revised version, we changed most of the names to be consistent with Dell’Aquila et al. The PHR and PSR both refer to the Myb

transcription factor, which share the same Myb DNA-binding domain with a SHLQKYR motif located at its C-terminal part^{4, 5, 6}. A SPX-PHR cascade regulation has been extensively studied in plants^{7, 8}. And in this study, in order to illustrate the similar cascade regulation, we still use PHR but with a note that it refers to the same gene as PSR (page 4, line 69).

(7) Lines 64-65, the statement is not accurate as Fig.1a is showing the alignment of only one of the P. tricornutum SPX domain containing genes, all the other proteins shown in figure are from plants. It would be more informative to add all the P. tricornutum sequences and maybe remove some of the plant sequences, as the focus is on the newly identified P. tricornutum genes.

Response to reviewer comment No.7: Changed as you suggested (see Fig. 1b in the revised manuscript).

(8) Line 67, supplementary figure 1, how were the sequences in the tree selected? There are plant sequences but also other diatom sequences, where do they come from? They must have been found in ncbi by blast, but no reference to this search can be found anywhere. Please add details.

Response to reviewer comment No.8: Added as you suggested. The phylogenetic tree was inferred based on the amino acid alignment of SPX domain-containing genes of different organisms. Each gene copy was BlastX against NCBI NR database, and representative sequences within the top 30 hits were selected for the phylogenetic analysis. The amino acid sequences were aligned by using MUSCLE⁹ embedded in the software MEGA X¹⁰, and edited manually using MEGA X. Phylogenetic tree was inferred employing the PhyML-aLRT method¹¹ with LG+G models and best of NNI & SPR on the Seaview platform¹².

(9) The authors obtained five SPX mutant lines, which one did they use for the analyses and which mutations were present in the selected mutant? What

was the effect of the mutations on the protein? Did they get a truncated protein, a protein with a missing part, or amino acid changes? It would be informative to visualize the position of the conserved sequence shown in 1a in the scheme in 1b, to show if it is before or after the targeting site of sgRNA C.

Response to reviewer comment No.9: In this study, five mutant strains of SPX (*mSPX*) were obtained. Our sequencing results showed that the two clones used for downstream analyses caused frameshift translation (*mSPX1*) and deletion of translation (*mSPX15*, lost 65 amino acids in the SPX domain), respectively. Furthermore, an expression analysis of SPX related genes was performed using qPCR on *mSPX* and WT grown under the P+ condition. The SPX expression in SPX mutants was significantly down-regulated compared with WT_P+ with a 4.4-fold and 6.2-fold decrease in *mSPX1* and *mSPX15*, respectively. So, *mSPX15* strain was used for the subsequent series of physiological and transcriptome analysis. Due to lack of suitable antibodies, we were unable to detect protein level changes in this study.

In addition, the position of the conserved sequence has been shown in the revised Figure 1.

(10) Is there a predicted localization for the P. tricornutum protein? Can they check with specific software?

Response to reviewer comment No.10: According to your suggestion, the subcellular protein location was predicted using the TargetP (<http://www.cbs.dtu.dk/services/TargetP>) server, showing that this SPX protein does not contain signal peptide, mitochondrial transfer peptide, chloroplast transfer peptide, and thylakoid luminal transfer peptide. It predicts this protein as a cytoplasmic protein.

(11) The presence of a truncated protein or of a mutated protein could lead to different effects with respect to a complete loss of the protein (function

partially retained, dominant negative effect, etc). There are Arabidopsis antibodies against SPXs, could the authors verify if they could work for this P. tricornutum SPX and possibly use them to check if a residual protein product is present? I think that the nature of the mutation in the protein could provide more information for the mechanism put forward in figure 4.

Response to reviewer comment No.11: We agree that with a specific antibody western blotting would provide more direct evidence of the mutated gene lost or drop its encoded protein in the cell. However, we could not get hold on a specific antibody. As an alternative, the mRNA expression analysis of SPX gene was performed using qPCR between *mSPX* (*mSPX1* and *mSPX15*) and WT under P+ condition. The SPX expression in *mSPX* was significantly down-regulated compared with WT_P+ showing a 4.4-fold and 6.2-fold decrease in *mSPX1* and *mSPX15*, respectively. These results also demonstrate the authenticity and reliability of the mutation.

(12) *Is there a growth phenotype in the mutant?*

Response to reviewer comment No.12: Yes, the *mSPX* strain grew slower than the WT strain (see page 6, lines 128-135)

(13) *The RNA-seq study and the qPCR analyses must be described better. Fig. 2c, d and e, at what time point were cell analyzed? Which phase of the growth curve was chosen? Were the qPCRs made on independent material or on the same RNAs used for the RNA-seq? Were other genes tested (and found not regulated) other than the ones shown in d and e? There is one gene in table S14 which does not appear in the paper.*

Response to reviewer comment No.13: We have now provided more information as advised (page 17, lines 470-489). Briefly here, the samples were collected on the third day during the early exponential growth period. The qPCRs, RNA-seq, and other physiological results were from the same time point (third day).

In the revised manuscript, an expression analysis in different mutant strains

(*mSPX1* and *mSPX15*) was performed using qPCR under P+ condition. The other two P-stress induced SPX domain-containing genes (*Vpt1* and *Vtc4*) were significantly up-regulated in both *mSPX1* and *mSPX15* compared to WT under P+ condition. In addition, three phospholipid degradation-related genes were tested using qPCR. Two of them displayed strongly induced expression (ranging from 2.3-fold to 4.5-fold up-regulation) both in *mSPX1* and *mSPX15* under P+ condition. While one phosphoethanolamine phospholipase (*ENTPPL*, *Phatr3_J52110*) showed up-regulation in RNA-seq, no significant difference was observed both in *mSPX1* and *mSPX15* by qPCR.

(14) Lines 90-97, a similar dataset was produced by Alipanah et al., 2018, <https://doi.org/10.1371/journal.pone.0193335>. I understand from the methods that the setup used here and in Alipanah were similar but not identical, anyway the purpose of finding genes involved in the P metabolism was the same, therefore I suggest to cite this study and comment on relevant differences (if any) in the discussion.

Response to reviewer comment No.14: Thank you for providing these two articles.

The two articles have been cited and fully discussed with our results in the revised manuscript. In addition, we revised the name of PTs according to Alipanah *et al.*, 2018 (doi.org/10.1371/journal.pone.0193335) and Sharma *et al.*, 2020 ([doi: 10.1111/nph.16248](https://doi.org/10.1111/nph.16248)).

(15) Line 97, change “rendering...” with “suggesting that the other five IPT could be...”

How are the high-affinity and low-affinity assignments made? Are there references to experimental data supporting the classification? I assume this is not based only on the gene expression profile. There is a tree in supplementary figure 2, some clades only contain diatom sequences (for instance the second cluster from the bottom), on which bases that clade is

defined as high-affinity? Again in this tree there are sequences from other diatoms but it is not explained how they were selected and where they come from. In fact, how were the other organisms used in tree selected?

Response to reviewer comment No.15: Since we realize that the PTs have been detailly described by Alipanah *et al.*, 2018 (doi.org/10.1371/journal.pone.0193335) and Sharma *et al.*, 2020 (doi: 10.1111/nph.16248), we directly followed their naming of these P transporters. In the revised manuscript, we removed the description of PT classification, and instead focus on the regulation of PTs in SPX mutant.

(16) Back to the RNA-seq, I wonder why the authors focus on subsets of genes in the various supplementary tables, which is fine, but do not provide general data on the results of the gene expression analysis, especially when involving the mutant. How many genes were differentially expressed in total between the mutant and the wt? Are there other major functions disrupted? Do the samples cluster correctly in the routine quality controls of transcriptomics experiments?

All these details should be given. I am also curious about whether any of the other SPX containing genes is regulated in the mutant.

Response to reviewer comment No.16: Initially we tried to keep the main paper short, but now as suggested by you we have added the details have in the revised manuscript. In addition, we also noted that the other two P-stress induced SPX domain-containing genes (Vpt1 and Vtc4) described in Gianluca *et al.* (doi: 10.3389/fpls.2020.00579), were significantly up-regulated in both *mSPX1* and *mSPX15* compared to WT under P+ condition. See page 5, lines 117-124.

(17) Lines 124-125, how were the PHR genes identified?

Response to reviewer comment No.17: Explanation is given in the revised manuscript. Based on the information that Myb transcription factors function as a central regulator of Pi starvation signaling in plants and algae⁶, we looked to

MybTFs when we suspected that an intermediate regulator likely occurred between SPX and the P acquisition machinery (e.g. P transporters and alkaline phosphatase). We examined the nine Myb TFs that contain a single Myb domain (Myb1R) described by Rayko *et al.*^{13, 14}. Among these, only Myb1R_5 (PHR) exhibited significant up-regulation under both P-stressed condition and in mutation condition (Supplementary Table 7), but showed no response in the recently generated PhoA and PhoD mutants¹⁵, indicating that it functions downstream of SPX and upstream of APs.

(18) To unequivocally demonstrate that the effects observed depend on the SPX mutation and not on background mutations in the mutant chosen for the RNA-seq, I would recommend to perform a few qPCRs on one of the other mutants to obtain further support to the specificity of the phenotype.

Response to reviewer comment No.18: Added as you suggested. In the revised manuscript, two mutant strains (*mSPX1* and *mSPX15*) were used for the qPCR test. Compared with WT, SPX expression dropped by 4.4-fold in *mSPX1* and 6.2-fold in *mSPX15*, both with statistical significance ($p < 0.05$). In addition, the other two SPX domain-containing genes (*Vpt1* and *Vtc4*), two APs, two PTs, and three phospholipid degradation-related genes were used for further expression analysis by qPCR. And the mRNA levels of these genes were similar between different mutant strains, so the possibility of background (off-target) mutations or damaging effects are eliminated.

(19) Line 140, the SPX and Myb domains and HMM profiles were used for the search, therefore what is found in the database are not only the orthologs to SPX (J47434) and PHR (J47256), but all the SPX and Myb containing genes, and some of these may have different functions. Without a more accurate analysis of the outputs obtained, it is not possible to claim that the two genes are expressed widely. The results simply indicate that the domains are broadly conserved.

Response to reviewer comment No.19: Thanks for the excellent insight. We totally agree, and now we add a description in Results (page 9, line 228-230) and a note in Discussion, last paragraph, “We should note that our search result using the conserved domain sequence as query likely includes SPX domain-containing multi-domain genes. Future research should use more gene type-specific tools to tease out how phylogenetically and geographically common each member of the SPX domain gene family is.

Regarding PHR, after considering the complexity of the Myb transcription factors, we decided to remove the global ocean search for PHR.

(20) Line 142, figure 3 is a screenshot of the output of the OGA blast, why are results shown for the deep chlorophyll maximum and not for the surface? In general, I think that screenshots are fine for a supplementary figure, but for a main figure I would try to provide something more informative. Can the general result in the Tara oceans dataset be summarized differently?

Response to reviewer comment No.20: Both SRF and DCM layer were displayed in the revised manuscript. In addition, the new summarized figure replaced the screenshots.

Discussion

(21) Since a KO is available for the other main partner in the model proposed (Sharma et al., 2020), the data should be cited and compared, and it should be discussed if they support the mechanism proposed in figure 4.

Response to reviewer comment No.21: Cited and discussed in the revised manuscript as you suggested.

(22) Line 169-170, the sentence is incorrect. Without measuring the lipids directly it cannot be claimed that phospholipid degradation is increased, there is only information on gene transcription in the paper.

Response to reviewer comment No.22: To remedy the issue, we have measured

neutral lipid content in *P. tricornutum* cells and added the results in the revised manuscript. Together with the transcriptional regulation of phospholipid degradation-related genes, the lipid synthesis in *P. tricornutum* was discussed fully in Discussion section.

Methods

(23) Lines 231-232, the rationale for the choice of one of the six SPX containing genes should be placed earlier in the text and possibly better detailed, the reason for excluding the other two induced genes is relevant information.

Response to reviewer comment No.23: Revised as you suggested. The question relates to the distinction of two groups of SPX domain-containing genes: a) genes containing SPX as the only functional domain, b) genes containing SPX domain and other functional domains. We chose to focus on type a to avoid interference of the additional domains when interpreting mutation data, and that was the only one type a gene in *P. tricornutum* genome.

(24) Line 293, I do not understand why *E. coli* had to be replated.

Response to reviewer comment No.24: Revised in the new manuscript. Firstly, the PCR products were purified, and cloned into pMD19-T plasmid that was transformed into DH5 α . Then, at least ten random clones were picked for sequencing to get an overview of the different indels.

(25) Line 335, the entire description of how differentially expressed genes were obtained is missing, which method/software/parameters were used? Please add details.

Response to reviewer comment No.25: Information added (page 19, lines 542-570).

(26) Line 342, which method was used to analyze the qPCR results? Please

add details.

Response to reviewer comment No.26: Information added. Initially, we used absolute quantitative qPCR (with known amount of target gene as standard) to calculate gene expression levels. And ribosomal protein small subunit 30S (RPS) were used as reference to calibrate the expression of related genes, because the expression level of RPS is relatively constant under different conditions^{1, 2, 3}. In the revised version, we decided to calculate relative quantitative qPCR to focus on the expression changes between *mSPX* and WT strains, and RPS was still used as the internal reference gene.

(27) Figure 2 and legend.

Why are the authors referring to “strains” in the legend? Is it a typo?

In this figure, if the qPCRs were made on the same material used for the RNA-seq (d) and (e) become less necessary (they confirm and repeat data found in tables) whereas it would be more interesting to see more of the general RNA-seq results. What is on the Y axis in d and e?

Response to reviewer comment No.27: The “strains” means the “mutant lines”. We feel it is important to verify transcriptomic results by running qPCR and therefore prefer that the result is presented. Furthermore, two mutant strains (*mSPX1* and *mSPX15*) in the revised manuscript were used here to verify the reliability of RNA-seq results and the authenticity of the mutation.

In addition, detailed RNA-seq analysis was conducted in the new version. The *mSPX* versus WT comparisons of gene expression under both P conditions revealed 885 differentially expressed genes (DEGs) (Supplementary Table 3). When analyzed separately, under the P+ condition, *mSPX* showed 528 DEGs (250 up-regulated and 278 down-regulated) relative to WT, and under the P- condition there were 560 DEGs (306 up-regulated and 254 down-regulated) (Supplementary Table 3). These revealed a dramatic impact of SPX mutation on the general transcriptomic landscape. (page 7, lines 170-176). Besides the P metabolism related genes, twelve fatty acid biosynthesis related genes

(Supplementary Table 10) (page 12, line 323) and seven ribosome biogenesis related genes (Supplementary Table 12) (page 13, lines 359-360) were significantly down-regulated and up-regulated in *mSPX* compared to WT, respectively.

The Y axis in d and e depicts the gene expression which was normalized to small subunit ribosomal protein 30 (RPS). RPS was chosen as reference because the expression level of RPS is relatively constant under different conditions^{1, 2, 3}. In the revised version, we changed to absolute qPCR to relative qPCR, which is sufficient to monitor expression changes between *mSPX* and WT strains, and RPS is still used as the internal reference gene.

Minor details

(28) Line 27, add an article before “phosphate”

Response to reviewer comment No.28: Changed as you suggested.

(29) Line 37, relies, not rely

Response to reviewer comment No.29: Changed as you suggested.

(30) Line 52, “its” referred to what? In the sentence before, SPX containing genes are mentioned.

Response to reviewer comment No.30: “its” referred to the SPX gene (Phatr3_J47434), which is now made clear. And detailed description has been added in the revised manuscript.

(31) Line 107, Eleven

Response to reviewer comment No.31: Changed as you suggested.

(32) Line 167, remove the dash in up-stream

Response to reviewer comment No.32: Changed as you suggested.

(33) Lines 351 and 352, HMM and hmm, should it be the same?

Response to reviewer comment No.33: Yes, they are the same.

(34) Line 511, add an article before “deletion”

Response to reviewer comment No.34: Changed as you suggested.

(35) Line 514, specify what the green color is

Response to reviewer comment No.35: Changed as you suggested.

(36) Line 516, add an article before “upstream”

Response to reviewer comment No.36: Changed as you suggested.

(37) Line 528, SPX genes

Response to reviewer comment No.37: Changed as you suggested.

(38) Line 536, I do not see any text underlined or italicized in the figure

Response to reviewer comment No.38: Corrected.

(39) Line 537, regulation functions?

Response to reviewer comment No.39: Changed as you suggested.

(40) Line 540, arrow indicates

Response to reviewer comment No.40: Changed as you suggested.

Reviewer #3 (Remarks to the Author):

*In this paper, the authors investigate the role of SPX (and SPX-related) proteins in phosphate homeostasis in algae, using the diatom *Phaodactylum tricorutum* as study case. According to what I have understood, the authors grew the alga in P-replete and P-limited conditions and then they produced transcriptomes in the same conditions. Such transcriptomes were used “to*

find” genes harbouring the SPX domains. Subsequently, they conducted a phylogenetic analysis to ascertain if such genes were homologous to the ones characterised in land plants. After positive results, they knocked out the SPX protein gene (Phatr3_J47434) and evaluated the effect of this alteration in the homeostasis of phosphate in mutant and wild type. Finally, using the metatranscriptomic data of Ocean Gene Atlas and the transcriptomes of MMETSP they searched for the presence of SPX genes across marine protists.

I think that this is an interesting paper contributing to the understanding of phosphate uptake and homeostasis in algae. The analyses are convincing of the results obtained but the way they are presented is confusing and difficult to follow. Indeed, after reading the manuscript several times, I still had difficulties in understanding some steps. For these reasons, I think that the manuscript needs some major revisions aiming at clarifying all the steps of the analyses and simplifying the reading before being considered for publication.

(1) For example, I am not sure on what samples the transcriptomes have been sequenced. From the read, it seems that the authors have produced transcriptomes from P-replete and P-limited grown algae to analyse the differential expression of genes in these conditions. However, I could not find any reference to this in the manuscript (or not in a clear way). Then, at lines 90 on, it seemed to me that the transcriptomes were performed on the wild types and mutant algae. I think this need to be stated clearly in the manuscript.

Response to reviewer comment No.1: Sorry for the confusion. We now have provided more information to clarify this. The short answer is that two sets of transcriptomes were sequenced: first on P+ and P- cultures to find SPX (page 15, lines 415-416), and the other on mutant and wild type to evaluate change of P related genes after mutation. Please see page 19, lines 542 to 556 for details.

(2) Related to this point, I wonder if the authors have also found other genes (known or not from literature) differentially expressed in P-replete and P-limited conditions worthy to be at least discussed (and not used for mutagenesis.

Response to reviewer comment No.2: According to your suggestion, the detailed results was added to the revised manuscript.

The *mSPX* versus WT comparisons of gene expression under both P conditions revealed 885 differentially expressed genes (DEGs) (Supplementary Table 3). When analyzed separately, under the P+ condition, *mSPX* showed 528 DEGs (250 up-regulated and 278 down-regulated) relative to WT, and under the P- condition there were 560 DEGs (306 up-regulated and 254 down-regulated) (Supplementary Table 3). These revealed a dramatic impact of SPX mutation on the general transcriptomic landscape (page 7, lines 170-176).

Besides the P metabolism related genes, twelve fatty acid biosynthesis related genes (Supplementary Table 10) (page 12, line 323) and seven ribosome biogenesis related genes (Supplementary Table 12) (page 13, lines 359-360) were significantly down-regulated and up-regulated in *mSPX* compared to WT, respectively.

(3) Another major concern I have is into the Introduction. It is too short and the status of knowledge of P uptake and homeostasis in photosynthetic organisms (land plants, protists, etc) is barely introduced. The authors say at line 45 that “very little is known about the machinery in marine phytoplankton” but do not say anything about the machinery in other photosynthesising organisms. There is no reference to the role of SPX proteins (as well as other proteins and signalling molecules) in phosphate homeostasis and to recent literature about it. I only found a reference to Rubio et al. (2001). Authors should include recent literature and explain the current model, even if it is proposed for land plants. This could also be of help

in the Discussion, since they conclude that a similar mechanism occurs in algae.

Response to reviewer comment No.3: Introduction has been elaborated with referencing to updated literature.

(4) Related to the aforementioned literature, I wonder if the authors have found a differential expression in their transcriptomes of MYB transcription factors and light-harvesting related (LHCSR) proteins.

Response to reviewer comment No.4: In this study, nine Myb TFs which contain a single Myb domain (Myb1R) described by Rayko *et al.*^{13, 14}, were investigated. Among these, only Myb1R_5 (PHR, Phatr3_J47256) exhibited significant up-regulation in *mSPX* compared to WT.

In addition, only one fucoxanthin chlorophyll a/c protein (Phatr3_J48882) was significantly up-regulated in *mSPX*, which occurred under P+ condition.

(5) The Discussion is very short and not written in the context of recent literature. Therefore, I believe that the merits of the findings are not properly highlighted and I strongly recommend to rewrite this section as well as the introduction.

Response to reviewer comment No.5: In the revision, we have more fully developed Discussion, with the expansion into six sections: SPX constraining phosphorus acquisition and recycling mechanisms, SPX constraining phosphorus storage, SPX constraining phospholipid degradation and neutral lipid synthesis, SPX-PHR regulatory cascade, Potential role of SPX in nitrogen homeostasis, and Potential universal presence and important role of SPX in global phytoplankton communities.

(6) The materials and methods are extensively written but sometimes the workflow is not clear. I am not an expert of the Crispr/Cas approach but still found difficult to understand.

Response to reviewer comment No.6: In the revision, we have elaborated, reorganized or rephrased in places where original narrative was too simplified, disconnected, or not clearly described. We also have checked the entire manuscript carefully to ensure it is clear.

SPECIFIC COMMENTS

(7) Line 22 = “we identify SPX protein”. The sentence appears without any reference before to SPX proteins/domains and their role. Furthermore, later on the authors say that they found several SPX or SPX-like proteins.

Response to reviewer comment No.7: We have revised it by introducing plant SPX-PHR first. It now reads: “Maintaining intracellular P homeostasis against environmental P variability is critical but how phytoplankton achieve it is poorly understood while a SPX-PHR regulatory mechanism is known in plants. Here we identify an SPX gene and investigate its role in *Phaeodactylum tricornutum*.”

(8) Lines 49-50 = it is not clear to me if the authors were already searching for SPX genes in the transcriptomes or if they wanted to first ascertain if they were differentially expressed in the two different P-availability conditions and then to analyse their role in detail.

Response to reviewer comment No.8: We have expanded Introduction now to explain the rationale and progressive flow of work. Briefly, based on the knowledge of the SPX-PHR regulatory mechanism in plants and the lack of equivalent information in phytoplankton, we attempted to find functional equivalent of this mechanism in the model diatom *Phaeodactylum tricornutum*.

(9) Line 61 = to what conditions the transcriptomes refer to?

Response to reviewer comment No.9: The transcriptomes used here were generated under P-limited condition. The information is now added (page 15, lines 415-416).

(10) Line 226 = here the authors state that they generated transcriptomes. Others? In what conditions? I think this is the point that confuses me.

Response to reviewer comment No.10: We are very sorry for the lack of clarity due to our oversight when changing the format of the manuscript before submission. We now have revised the whole Methods section to make things clear. Briefly, the SPX domain-containing genes were identified from the genome. Their identity as SPX was verified by a phylogenetic tree inference. Then, their expression dynamics under P+ and P- conditions was examined using P+ and P- transcriptomes. From these data, SPX was selected for knockout study. Throughout the study, the SPX mutant transcriptomes were also generated for P+ and P- culture conditions. From these total of 12 transcriptomes, gene expression profiles in wild type and in SPX mutant, grown under P+ and P- were analyzed.

(11) Lines 283-285 = the concept is not clear. Please explain it better.

Response to reviewer comment No.11: We have revised these sentences to make it clear. See page 16, lines 456-458. To recap, for checking whether the Cas9 component was successfully transformed into the cells, primer pairs were designed in the 3' region of Cas9 sequence for use in PCR to amplify the Cas9 fragment.

(12) Line 334 = I do not understand what the authors mean for “quantify expression levels”. From the mapping procedure? It sounds strange to me. Then after a few lines, the authors state that they have used RT-qPCR for quantifying the expression levels. Furthermore, for such analysis there is no specification of what the reference genes are and how the data normalisation has been performed.

Response to reviewer comment No.12: This part has been revised and information needed. Main information can be seen as follows:

After sequencing and quality filtering, the reads were mapped to the species'

genome, and StringTie and DESeq2 software were used to estimate transcript abundances and identify differentially expressed genes between mutant and WT groups, respectively¹⁶. Genes with FDR < 0.05 (adjusted p-value) and log₂ Fold Change > 1 were used to define as statistically significant DEGs in this study.

For the RT-qPCR, ribosomal protein small subunit 30S (RPS) were used as reference to calibrate the expression of related genes, because the expression level of RPS is relatively constant under different conditions^{1,2,3}. In the revised version, we changed to relative quantitative qPCR to reflect the expression changes between *mSPX* and WT strains, and RPS still used as the internal reference gene.

(13) Line 349. The authors have also searched for MYB domain and PHR proteins. Indeed, MYB transcription factor is known to be involved in phosphate starvation signalling in vascular plants and unicellular algae (see Rubio et al. 2001) but there is no reference to this in the Introduction.

Response to reviewer comment No.13: Reference added.

References

1. Baïet B, *et al.* N-glycans of *Phaeodactylum tricornutum* diatom and functional characterization of its N-acetylglucosaminyltransferase I enzyme. *J Biol Chem* **286**, 6152-6164 (2011).
2. Bailleul B, *et al.* An atypical member of the light-harvesting complex stress-related protein family modulates diatom responses to light. *Proc Natl Acad Sci U S A* **107**, 18214-18219 (2010).
3. Saade A, Bowler C. Molecular tools for discovering the secrets of diatoms. *BioScience* **59**, 757-765 (2009).
4. Wykoff DD, Grossman AR, Weeks DP, Usuda H, Shimogawara KJPotNAoS. Psr1, a nuclear localized protein that regulates phosphorus metabolism in *Chlamydomonas*. **96**, 15336-15341 (1999).
5. Thiriet-Rupert S, *et al.* Transcription factors in microalgae: genome-wide prediction and comparative analysis. **17**, 1-16 (2016).
6. Rubio V, *et al.* A conserved MYB transcription factor involved in phosphate starvation signaling both in vascular plants and in unicellular algae. *Genes Dev* **15**, 2122-2133 (2001).
7. Lv Q, *et al.* SPX4 negatively regulates phosphate signaling and homeostasis through its interaction with PHR2 in Rice. *Plant Cell* **26**, 1586-1597 (2014).
8. Puga MI, *et al.* SPX1 is a phosphate-dependent inhibitor of Phosphate Starvation Response1 in *Arabidopsis*. *Proc Natl Acad Sci U S A* **111**, 14947-14952 (2014).
9. Edgar RC. MUSCLE: multiple sequence alignment with high accuracy and high throughput. *Nucleic Acids Res* **32**, 1792-1797 (2004).
10. Kumar S, Stecher G, Li M, Knyaz C, Tamura K. MEGA X: Molecular evolutionary genetics analysis across computing platforms. *Molecular Biology and Evolution* **35**, 1547-1549 (2018).
11. Guindon S, Dufayard JF, Lefort V, Anisimova M, Hordijk W, Gascuel O. New algorithms and methods to estimate maximum-likelihood phylogenies:

- assessing the performance of PhyML 3.0. *Syst Biol* **59**, 307-321 (2010).
12. Gouy M, Guindon S, Gascuel O. SeaView version 4: A multiplatform graphical user interface for sequence alignment and phylogenetic tree building. *Mol Biol Evol* **27**, 221-224 (2010).
 13. Rayko E, Maumus F, Maheswari U, Jabbari K, Bowler C. Transcription factor families inferred from genome sequences of photosynthetic stramenopiles. *New Phytol* **188**, 52-66 (2010).
 14. Sharma AK, *et al.* The Myb-like transcription factor phosphorus starvation response (PtPSR) controls conditional P acquisition and remodelling in marine microalgae. *New Phytol* **225**, 2380-2395 (2020).
 15. Zhang K, *et al.* One enzyme many faces: alkaline phosphatase-based phosphorus-nutrient strategies and the regulatory cascade revealed by CRISPR/Cas9 gene knockout. *bioRxiv*, (2020).
 16. Love MI, Huber W, Anders S. Moderated estimation of fold change and dispersion for RNA-seq data with DESeq2. *Genome Biol* **15**, 550 (2014).

Reviewers' Comments:

Reviewer #1:

Remarks to the Author:

I think that the manuscript has improved by the revision process. The methodological part is now more detailed, additional analyses have been done, physiological results have been added, and supplementary datasets have been used that made the results and conclusions more robust. The authors have addressed many of the comments suggested by the three reviewers. However, not all questions/comments from the first round of reviews have been addressed. In addition, a number of inconsistencies in the text and the figures make this manuscript difficult to follow.

I have noticed a number of discrepancies between different parts of the text or between the text and the figures which made it rather difficult and time consuming to follow the storyline of the article. What's even more important, these discrepancies made it difficult to appreciate the importance of the presented results. I have now read this article three times and I still struggled, which is, in my opinion, an indication that it may be difficult for the reader to understand the importance of this work.

To be able to follow the article, to understand the importance of its results, and to reproduce computational analyses, all inconsistencies should be eliminated. For the reproducibility of the results it is especially important to know what version and what parameters for each software package were used. Even more so, if not default parameters were used.

It was noted and suggested by Reviewer #2 in comment 25 that "the entire description of how differentially expressed genes were obtained is missing, which method/software/parameters were used? Please add details."

Only some of this information has been provided in the revised version of the manuscript and not where it was indicated in the rebuttal letter (page 19, lines 542-570) - probably not the latest version of the rebuttal letter OR the manuscript has been provided for the revision, which made the review process more cumbersome.

I couldn't find a reply to some of my questions in comment 7. More specifically, "How did taxonomic diversity vary across the size fractions, were the same organisms responsible for the expression of these genes or did different organisms dominate the community depending on the SF?". In fig 4b the authors present "Phylogenetic map showing the taxonomic diversity of SPX domain-containing unigenes in all samples." Which is fine and probably sufficient to be presented as one of the main figures in the paper. Nevertheless, did taxonomic diversity vary across the size fractions or did it remain the same? If it didn't vary, it's sufficient to just add this short sentence to the text and use fig 4b as is. And if it did vary, describe how.

Comments:

Line 83-86. It would probably be better to separate these datasets in such a way so that it's clear that MMETSP is a collection of reference transcriptomes derived from cultured organisms, MATOU is a catalog of genes derived from the Tara Oceans expedition and the MGT collection is a collection of metagenomics based transcriptomes derived from the MATOU catalog.

Line 84. "which aligned well" - Please define "well", what % over what length of the sequence or give some other metric

Line 101. "were differentially expressed" - Please provide what metric was used, what p value, what log₂FC?

Line 108. "all on the third of our three designated targets" - I don't understand this sentence

Line 113. "clone 15-3 or mSPX15" - If there's no specific reason to do so, please use the same name for each mutant throughout the text and in ALL the figures. It appears that in fig 1 different names are used for the same mutants on different panels. It makes it more difficult to follow the story.

Lines 118-122. I'm not sure I follow. I can't find the described here genes in Fig 1g.

Line 171. "revealed 885 differentially expressed genes" - There are almost 1100 genes in table S3, please clarify.

Lines 170-175. I think a further clarification is needed, either here or in the M&M section.

As of now it says in the M&M section "Genes with FDR < 0.05 (adjusted p-value) and log₂ Fold Change > 1 were defined as DEGs. In total two different comparisons were analyzed in our study: DEGs in mSPX_P+/WT_P+ comparison to explore the role of SPX in the P-replete condition, and DEGs in mSPX_P-/WT_P- comparison to explore the SPX function in the P-deficient condition."

So the conditions were analyzed both separately and together? If so, please describe what parameters of the DESeq2 pipeline were used to run these analyses. In fact, an R Markdown file or a similar code report file would be useful in the Suppl Mat.

Line 189 "81.6-fold upregulation" - FC is used in the text and log₂FC is used in the table. It is difficult to follow.

Line 191 "Fig. 3c" - What do numbers on the x axis mean? All qPCR related figures are somewhat hard to follow because I can't match the names in the text with the names in the figures.

Line 197 "both P+ and P- conditions" - Not all 11 genes were significantly upregulated under both P+ and P- conditions, please reflect that in the text.

Line 200 "2.3-fold to 4.5-fold upregulation" - Again, fold change is presented in the text, but log₂FC is presented in the figure. And the names mentioned in the text (hatr3_J32057 and Phatr3_J49693) are not present in the figure. Please address this issue for all the figures.

Line 235 "unigenes" - How do you define a unigene? This term is used in some of the Tara Oceans studies but how do you define it here, including in table s3?

Line 567 - Please provide parameters that were used for the DESeq2 analysis. Also, if not default parameters were used, indicate what statistical test was used for hypothesis testing when comparing two groups (Wald test is default)

Line 567 - Different terms are used to describe the adjusted p value cutoff throughout the paper, adjusted p value, q value, and FDR. Please keep it consistent throughout the manuscript AND the figures or explain why different terms are used.

Line 590 - Please indicate these units (μm) in the relevant figure

Line 851 - g. What do numbers on the x axis mean?

Line 866 - I don't understand what different letters indicate. Significant difference at different p values? If yes, which p values? Or something else? Please clarify.

Typos:

Lines 53-54. It should probably read: Additionally, we find that the SPX related genes exist and operate across phytoplankton phylogenetic spectrum and global oceans, indicating its universal

importance in marine phytoplankton.

Line 255. "is" is missing

Line 261. indicates

Reviewer #2:

Remarks to the Author:

The manuscript has indeed improved, and I am happy to see that changes and additions have been made in reply to my comments.

Regarding reply #11, if antibodies are not available of course no protein study is doable, and I accept this. The qPCR data provided as an alternative (Fig 1g) are interesting. Not necessarily gene expression is affected when a mutation is introduced in the coding sequence. In principle, the gene could be normally expressed and a mutated protein could be produced in amounts equal to the wild type. The biological effect is normally due to the defective protein, not to the reduced levels of mRNA. The issue should be downstream of transcription. However, I think there are reports of CRISPR/Cas9 mutations having an unexpected effect on the mRNA levels, therefore from the data it seems like there could be additional mechanisms at play (nonsense-mediated mRNA decay?) in this diatom too. I suggest that the authors look at the relevant literature and evaluate whether it is appropriate to add a comment in the manuscript.

A doubt about Fig. 1d, the presence of two bands in mSPX_15 indicates that one allele is mutated (deletion, the lower band) and the other is wild type? Has the top band of a height similar to the wild type been sequenced? It seems like it is wild type but it might also have small changes not detectable on the gel.

I think that the English still needs revision. In a few places I would add or remove articles, I suggest that the manuscript is proofread by a native speaker.

Below some other minor issues I still found

Lines 30-31 in the abstract, check noun verb agreement.

Line 71, replace "that the PHR protein acts" with "to act"

Line 91, I would write "P. tricorutum genome" (instead of existing genome)

Line 93, Gianluca is the first name, the citation should be Dell'Aquila et al.

Line 119, add "the" before strongest

Line 255-256, the verb is missing

Line 263, might exist

Linea 362-363, is there a reference to support the link between ribosome biogenesis and elevated nitrogen metabolism?

Reviewer #3:

Remarks to the Author:

This revised version of the manuscript has greatly benefitted from the suggestions of all reviewers. I appreciate the extensive work that the authors have done to address the points raised by me and the other reviewers, which resulted in some sections (introduction, M&M, discussion) greatly improved. Overall, the manuscript has now a well-defined structure and it more readable than the previous version. The discussion of results in the context of existing data is now satisfying and extensive. However, there are still some points that I would like to see fixed in this version, and I hope that the authors will be willing to do this last effort.

In the Results section, for example, I think that the description of transcriptome analyses should be placed before the Crispr/Cas experiment and after the blast search of SPX genes in the *P. tricornutum* genome. Indeed, at lines 102-103 the authors refer to differential expression of SPX, Vpt1 and Vtc4 genes (that were then used for mutagenesis) without reporting the results of transcriptome analyses. This "patchy" way of reporting results makes difficult the reading of the manuscript. To solve this, they could report the results in the same order of the M&M, where P+ and P- transcriptomes have been placed after blast analysis. I am aware that several, different approaches have been used in this study, and that they have performed differential expression analyses on both WT and mutants; However, I believe that it would be better to split the results according to a workflow rather than techniques to guide the reader.

SPECIFIC COMMENTS

Line 38 = contributes for about 50% to global primary production

Line 41 = please replace "P-nutrient" with "phosphorus" or "P" and then reformulate as follows: "primarily in the form of dissolved..."

Please refer to phytoplankton as a singular noun throughout the manuscript; otherwise refer to phytoplankton species or organisms

Line 66 = within phytoplankton, the species in which P regulation has been best studied is...

Line 68-69 = already stated in the introduction. Please remove "and it is a species of diatoms, a dominant group of phytoplankton contributing about 40% of marine primary production"

Line 70-71 = "has been identified by gene knockout experiments, with the PHR protein acting as a positive regulator of Pi signalling"

Line 86 = to examine the distribution and expression of the SPX. What do you mean here?

Line 93 = please replace "Gianluca" with Dell'Aquila et al.

Line 94 = "verified the presence" please replace with "allowed the identification"

Line 95 = you can shorten the sentence stating that the sequences of *P. tricornutum* shared several conserved sites with land plants' SPX domains

Line 96 = it is not "affiliation", you may want to use phylogenetic closeness, similarity

Line 256 = non-sense sentence. I believe there is a missing verb

Line 262 = the subject is mining, therefore the verb cannot be "indicate". "Furthermore, mining of the TARA Oceans metatranscriptomic data, we clearly found evidence that this SPX-based P-homeostasis regulatory mechanism exists..."

Line 264 = oceans

Line 387 = Guillard's F/2 medium. Add also the reference: Guillard, R. R. L. Culture of Phytoplankton for Feeding Marine Invertebrates. in Culture of Marine Invertebrate Animals: Proceedings — 1st Conference on Culture of Marine Invertebrate Animals Greenport (eds. Smith, W. L. & Chanley, M. H.) 29-60, https://doi.org/10.1007/978-1-4615-8714-9_3 (Springer US, 1975).

Line 392 = except that in P treatments, where phosphate concentrations provided in the media varied

Line 395 = "a P+ culture" – a P-replete culture?

Line 410 -412 = please put this sentence soon after the one above and add a new paragraph line after. If I have properly understood, at lines 405-412 you are summarising the protocol and from 413 onward you are providing details.

At this point, after selecting putative homologs of SPX by genome blast, why the authors did not also blasted this sequence against MMTSP transcriptomes and Tara/MGT metatranscriptomes and infer a phylogenetic tree with all these sequences? This approach could have been of help in ascertaining the identity of the sequence.

Line 412 = please specify at what similarity threshold (e-value) the search was performed.

Line 407 = please specify the Pfam number used to allow reproducibility

Line 415 = please revise this sentence. The aLRT (approximate likelihood-ratio test) is a test to perform the support to branches. The default option for PhyML is the Shimodaira-Hasegawa (SH) like test. Did you perform this test? Please also make clear that you have inferred a maximum likelihood phylogenetic tree in PhyML, that was ran as daughter process of SeaView (correct the name in the text).

Line 418-419 = "expression dynamics of the identified SPX domain-containing genes were examined from transcriptomes sequenced under P+ and P- conditions". I do not understand the meaning of this sentence with a reference here. Do the authors mean that they have analysed the expression profiles of previously published transcriptomes? Why the reference? If so, they should argue better the reasons for this choice. If not, they should rephrase the sentence because it is not clear.

Line 434 = at the beginning of M&M section there already is a paragraph about culture conditions. What experiments is it referring to? This is not clear. You may want to make a single paragraph on culture conditions for all the experiments instead of repeating it.

Line 840 = conserved sites

Figure 4b = better to replace "phylogenetic maps...." with "Krona pie-charts showing the taxonomic distribution of SPX-domain containing genes"

Response to reviewer comments (*italics*) for

“A regulatory cascade of phosphorus homeostasis in marine phytoplankton”, which was reviewed for **Communications Biology**.

Reviewer #1 (Remarks to the Author):

I think that the manuscript has improved by the revision process. The methodological part is now more detailed, additional analyses have been done, physiological results have been added, and supplementary datasets have been used that made the results and conclusions more robust. The authors have addressed many of the comments suggested by the three reviewers. However, not all questions/comments from the first round of reviews have been addressed. In addition, a number of inconsistencies in the text and the figures make this manuscript difficult to follow.

I have noticed a number of discrepancies between different parts of the text or between the text and the figures which made it rather difficult and time consuming to follow the storyline of the article. What's even more important, these discrepancies made it difficult to appreciate the importance of the presented results. I have now read this article three times and I still struggled, which is, in my opinion, an indication that it may be difficult for the reader to understand the importance of this work.

To be able to follow the article, to understand the importance of its results, and to reproduce computational analyses, all inconsistencies should be eliminated. For the reproducibility of the results, it is especially important to know what version and what parameters for each software package were used. Even more so, if not default parameters were used.

Response: We appreciate the general comments, which guided us to further improve the manuscript. We have carefully addressed them in the new version.

(1) It was noted and suggested by Reviewer #2 in comment 25 that “the entire description of how differentially expressed genes were obtained is missing, which method/software/parameters were used? Please add details.”

Only some of this information has been provided in the revised version of the manuscript and not where it was indicated in the rebuttal letter (page 19, lines 542-570) - probably not the latest version of the rebuttal letter OR the manuscript has been provided for the revision, which made the review process more cumbersome.

Response to reviewer comment No.1: Thank you for your suggestion. The detailed information has been added to the manuscript (see page 22, lines 626-645).

(2) I couldn't find a reply to some of my questions in comment 7. More specifically, “How did taxonomic diversity vary across the size fractions, were the same organisms responsible for the expression of these genes or did different organisms dominate the community depending on the SF?”. In fig 4b the authors present “Phylogenetic map showing the taxonomic diversity of SPX domain-containing unigenes in all samples.” Which is fine and probably sufficient to be presented as one of the main figures in the paper. Nevertheless, did taxonomic diversity vary across the size fractions or did it remain the same? If it didn't vary, it's sufficient to just add this short sentence to the text and use fig 4b as is. And if it did vary, describe how.

Response to reviewer comment No.2: Our main goal of analyzing SPX in the global ocean was to determine if SPX occurs widely geographically and taxonomically in general. Taxonomic similarity or dissimilarity among the size fractions are not immediately relevant to this goal. Nevertheless, we think the information is certainly beneficial and we did analyze it in this round of revision. The pico-eukaryote size fraction (0.8-5 μm) was dominated by Chlorophyta in subsurface layer (SRF) and Fungi in deep chlorophyll maximum layer (DCM), respectively (Supplementary Fig. 2, newly added and shown below). In contrast, the other size fractions were dominated by Bacillariophyta (5-20 μm) and Metazoa (20-180 μm ,

and 180-2000 μm) in both SRF layer and DCM layer, respectively (see page 10, lines 268-273). Based on the above analysis, taxonomic diversity varied across size fractions.

Figure 1 (Fig. S2 in the text) Expression of the SPX domain-containing genes in different lineages and different size fractions. The expression values were computed as RPKM. SRF, subsurface; DCM, deep chlorophyll maximum layer. The SRF layer and DCM layer were displayed on the left and right, respectively. The color block depicts the expression of certain lineage in different size fractions (0.8-5 μm , 5-20 μm , 20-180 μm , and 180-2000 μm).

Comments:

(3) Line 83-86. It would probably be better to separate these datasets in such a way so that it's clear that MMETSP is a collection of reference transcriptomes derived from cultured organisms, MATOU is a catalog of genes derived from the Tara Oceans expedition and the MGT collection is a collection of metagenomics based transcriptomes derived from the MATOU catalog.

Response to reviewer comment No.3: Separated as you suggested.

(4) Line 84. "which aligned well" - Please define "well", what % over what

length of the sequence or give some other metric.

Response to reviewer comment No.4: To make it clear and concise, we have revised this sentence to “Pfam analysis of the six identified sequences in *P. tricornutum* allowed the identification of SPX domain in these proteins, which shared several conserved sites with land plants’ SPX domains” (see page 5, lines 101-104).

(5) Line 101. “were differentially expressed” - Please provide what metric was used, what p value, what log2FC?

Response to reviewer comment No.5: The differentially expressed genes were identified from our RNA-seq database with DESeq2 (parameters: log2 Fold Change > 1 and adjusted p-value < 0.05)¹. The Wald test was used to calculate p-values, and then the p-values were adjusted for multiple testing using the Benjamini and Hochberg procedure. In the revised Table S1, the significant changes based on these criteria (log2 FC > 1 and adjusted p-value < 0.05) were indicated by asterisk.

(6) Line 108. “all on the third of our three designated targets” - I don’t understand this sentence.

Response to reviewer comment No.6: Totally, we designed three target sites, and the one that resulted in desired mutation happened to the third one. Therefore, in this manuscript, we designated it as sgRNA3. In order to avoid confusing readers, we deleted this sentence in the new version.

(7) Line 113. “clone 15-3 or mSPX15” - If there’s no specific reason to do so, please use the same name for each mutant throughout the text and in ALL the figures. It appears that in fig 1 different names are used for the same mutants on different panels. It makes it more difficult to follow the story.

Response to reviewer comment No.7: We have checked and made sure for our focused mutant *mSPX15-3*, we now consistently refer to it as *mSPX15-3*. In some places we used two mutants in the experiment and we mentioned both (the other

being *mSPX1-4*).

(8) Lines 118-122. I'm not sure I follow. I can't find the described here genes in Fig 1g.

Response to reviewer comment No.8: The three numbers (47434, 50019, and 19586) correspond to SPX, Vtc4, and Vpt1, respectively. In the revised Figure 1g, the gene names (SPX, Vtc4, and Vpt1) replaced the numbers to make the reading clearer.

(9) Line 171. "revealed 885 differentially expressed genes" - There are almost 1100 genes in table S3, please clarify.

Response to reviewer comment No.9: Sorry for not having made this clearer. Table S3 showed 1088 DEGs out of two kinds of comparison. Because 203 of these DEGs appeared in both comparisons, these DEGs represent 885 differentially expressed unigenes. We have added this explanation in the title of Table S3 and the main text.

(10) Lines 170-175. I think a further clarification is needed, either here or in the M&M section.

Response to reviewer comment No.10: We have elaborated slightly to explain how the number of DEGs were derived and what they represent (see page 8, lines 191-202; page 22, lines 633-645).

(11) As of now it says in the M&M section "Genes with $FDR < 0.05$ (adjusted p -value) and \log_2 Fold Change > 1 were defined as DEGs. In total two different comparisons were analyzed in our study: DEGs in *mSPX_{P+}/WT_{P+}* comparison to explore the role of SPX in the P-replete condition, and DEGs in *mSPX_{P-}/WT_{P-}* comparison to explore the SPX function in the P-deficient condition."

So the conditions were analyzed both separately and together? If so, please describe what parameters of the DESeq2 pipeline were used to run these

analyses. In fact, an R Markdown file or a similar code report file would be useful in the Suppl Mat.

Response to reviewer comment No.11: Information has now been added in the M&M section.

In this study, we detect DEGs using the DESeq2 package (<http://www.bioconductor.org/packages/release/bioc/html/DESeq2.html>) as requested. DESeq2 is based on the negative binomial distribution, performed as described at Love, *et al.*¹. The DEGs in different comparisons were analyzed separately, but the same parameters (log₂ Fold Change > 1 and adjusted p-value < 0.05) were used. In the new manuscript, a detailed description was added to this part, including the parameters and download addresses of the packages (see page 22, lines 633-645).

(12) Line 189 “81.6-fold upregulation” - FC is used in the text and log₂FC is used in the table. It is difficult to follow.

Response to reviewer comment No.12: As answered in comment 11, the parameter of log₂ Fold Change > 1 was used in DEGs analysis. Therefore, in order to maintain consistency, the expression regulation in RNA-seq tables was still displayed with log₂ Fold Change. That is easy for determining if the change is significant. However, in the text, it is easier to understand the magnitude of change when real fold changes are shown.

(13) Line 191 “Fig. 3c” - What do numbers on the x axis mean? All qPCR related figures are somewhat hard to follow because I can’t match the names in the text with the names in the figures.

Response to reviewer comment No.13: The numbers are the abbreviated Gene ID in *P. tricornutum* genome annotation database. In the new manuscript, we have revised them to full gene ID in Fig. 3c.

(14) Line 197 “both P+ and P- conditions” - Not all 11 genes were significantly upregulated under both P+ and P- conditions, please reflect that in the text.

Response to reviewer comment No.14: Revised as you suggested.

(15) Line 200 “2.3-fold to 4.5-fold upregulation” - Again, fold change is presented in the text, but log₂FC is presented in the figure. And the names mentioned in the text (*hatr3_J32057* and *Phatr3_J49693*) are not present in the figure. Please address this issue for all the figures.

Response to reviewer comment No.15: In order to maintain consistency with RNA-seq results, we think that using log₂FC is better to show on plots (which also makes it easier to present data of variable fold changes in same plot), but in text straight fold change is easier for readers to understand the magnitude of change. Titles on the y axis have been revised from “Fold change (log₂)” to “log₂ (fold change)” in both Fig. 3c and Fig. 1g to be clearer. In addition, we also deleted the description of “2.3-fold to 4.5-fold upregulation”.

The gene names (*Phatr3_J32057* and *Phatr3_J49693*) have been added to Fig. 3c in the revised manuscript.

(16) Line 235 “unigenes” - How do you define a unigene? This term is used in some of the Tara Oceans studies but how do you define it here, including in table s3?

Response to reviewer comment No.16: We did not define this, but only followed the name of the MATOU-v1 catalog. We now think that the reviewer brought up a good point that our use of the term “unigene” here can be misleading. Therefore, we have replaced “unigenes” with “genes” and made clear that the MATOU-v1 is a Unigene database.

(17) Line 567 - Please provide parameters that were used for the DESeq2 analysis. Also, if not default parameters were used, indicate what statistical test was used for hypothesis testing when comparing two groups (Wald test is default)

Response to reviewer comment No.17: In this study, the parameters (log₂ Fold

Change > 1 and adjusted p-value < 0.05) were used. The Wald test in this study was used to calculate p-values, and then the p-values were adjusted for multiple testing using the procedure of Benjamini and Hochberg. The information has been added (See page 22, lines 633-641).

(18) Line 567 - Different terms are used to describe the adjusted p value cutoff throughout the paper, adjusted p value, q value, and FDR. Please keep it consistent throughout the manuscript AND the figures or explain why different terms are used.

Response to reviewer comment No.18: In the new manuscript, the “q value” and “FDR” have been revised to “adjusted p value” in the text and supplementary tables.

(19) Line 590 - Please indicate these units (μm) in the relevant figure

Response to reviewer comment No.19: Revised as you suggested.

(20) Line 851 - g. What do numbers on the x axis mean?

Response to reviewer comment No.20: The numbers is the abbreviated Gene ID in *P. tricornutum*. In the new manuscript, we have revised them to full ID.

(21) Line 866 - I don't understand what different letters indicate. Significant difference at different p values? If yes, which p values? Or something else? Please clarify.

Response to reviewer comment No.21: Statistically significant differences are indicated by different letters, while the same letter means no significant difference. In the revised manuscript, asterisks were used to indicate a significant difference, * $P < 0.05$ and ** $P < 0.01$, ANOVA test (see page 32, line 909; Fig. 2).

Typos:

(22) Lines 53-54. It should probably read: Additionally, we find that the SPX related genes exist and operate across phytoplankton phylogenetic spectrum

and global oceans, indicating its universal importance in marine phytoplankton.

Response to reviewer comment No.22: Changed as you suggested.

(23) Line 255. "is" is missing

Response to reviewer comment No.23: Corrected.

(24) Line 261. indicates

Response to reviewer comment No.24: Changed as you suggested.

Reviewer #2 (Remarks to the Author):

The manuscript has indeed improved, and I am happy to see that changes and additions have been made in reply to my comments.

(1) Regarding reply #11, if antibodies are not available of course no protein study is doable, and I accept this. The qPCR data provided as an alternative (Fig 1g) are interesting. Not necessarily gene expression is affected when a mutation is introduced in the coding sequence. In principle, the gene could be normally expressed and a mutated protein could be produced in amounts equal to the wild type. The biological effect is normally due to the defective protein, not to the reduced levels of mRNA. The issue should be downstream of transcription. However, I think there are reports of CRISPR/Cas9 mutations having an unexpected effect on the mRNA levels, therefore from the data it seems like there could be additional mechanisms at play (nonsense-mediated mRNA decay?) in this diatom too. I suggest that the authors look at the relevant literature and evaluate whether it is appropriate to add a comment in the manuscript.

Response to reviewer comment No.1: Thanks for this insightful comment. We double checked and found the reason. Our qPCR primers (F1/R1) were located

near the mutation site (Fig. 1c), where the bases changed as a result of mutation, which would cause a decrease in amplification efficiency. To verify this hypothesis, we designed another pair of primers (F2/R2) on the 3' end, which was far away from the mutation site (Fig. 1c). The qPCR analysis using the 3' primers showed that the expression of SPX exhibited no significant change compared to WT (Fig. 1g). Also consistent with your comment, from our RNA-seq data, the expression of SPX in *mSPX* exhibited no significant changes compared with that in WT. We have provided this information in the revised manuscript.

(2) A doubt about Fig. 1d, the presence of two bands in mSPX_15 indicates that one allele is mutated (deletion, the lower band) and the other is wild type? Has the top band of a height similar to the wild type been sequenced? It seems like it is wild type but it might also have small changes not detectable on the gel.

Response to reviewer comment No.2: Both were mutated sequences of SPX. *P. tricornutum* cells could induce non-homologous recombination repair after the CRISPR/Cas9 mutation and multiple repair phenotypes usually exist in the first generation strain². We sequenced the top band (as well as the lower band that we focused on in this study), and found that small insertions and deletions (indels) occurred.

(3) I think that the English still needs revision. In a few places I would add or remove articles, I suggest that the manuscript is proofread by a native speaker.

Response to reviewer comment No.3: The revised manuscript has been proofread by Dr. Brittany Sprecher, who is an English native speaker and a marine phytoplankton biologist.

Minor issues

(4) Lines 30-31 in the abstract, check noun verb agreement.

Response to reviewer comment No.4: Changed as you suggested.

(5) Line 71, replace “that the PHR protein acts” with “to act”

Response to reviewer comment No.5: Changed as you suggested.

(6) Line 91, I would write “P. tricornutum genome” (instead of existing genome)

Response to reviewer comment No.6: Changed as you suggested.

(7) Line 93, Gianluca is the first name, the citation should be Dell’Aquila et al.

Response to reviewer comment No.7: Changed as you suggested.

(8) Line 119, add “the” before strongest

Response to reviewer comment No.8: Changed as you suggested.

(9) Line 255-256, the verb is missing

Response to reviewer comment No.9: Changed as you suggested.

(10) Line 263, might exist

Response to reviewer comment No.10: Changed as you suggested.

(11) Line 362-363, is there a reference to support the link between ribosome biogenesis and elevated nitrogen metabolism?

Response to reviewer comment No.11: Added as you suggested.

Reviewer #3 (Remarks to the Author):

This revised version of the manuscript has greatly benefitted from the suggestions of all reviewers. I appreciate the extensive work that the authors have done to address the points raised by me and the other reviewers, which resulted in some sections (introduction, M&M, discussion) greatly improved. Overall, the manuscript has now a well-defined structure and it more readable than the previous version. The discussion of results in the context of existing data is now satisfying and extensive. However, there are still some points that

I would like to see fixed in this version, and I hope that the authors will be willing to do this last effort.

(1) In the Results section, for example, I think that the description of transcriptome analyses should be placed before the Crispr/Cas experiment and after the blast search of SPX genes in the P. tricornutum genome. Indeed, at lines 102-103 the authors refer to differential expression of SPX, Vpt1 and Vtc4 genes (that were then used for mutagenesis) without reporting the results of transcriptome analyses. This “patchy” way of reporting results makes difficult the reading of the manuscript. To solve this, they could report the results in the same order of the M&M, where P+ and P- transcriptomes have been placed after blast analysis. I am aware that several, different approaches have been used in this study, and that they have performed differential expression analyses on both WT and mutants; However, I believe that it would be better to split the results according to a workflow rather than techniques to guide the reader.

Response to reviewer comment No.1: The transcriptome data used in this gene search effort have recently been published and we now cited the source (page 5, lines 109-112). These data are different from the transcriptome data presented in the later part of the manuscript.

SPECIFIC COMMENTS

(2) Line 38 = contributes for about 50% to global primary production

Response to reviewer comment No.2: Changed as you suggested.

(3) Line 41 = please replace “P-nutrient” with “phosphorus” or “P” and then reformulate as follows: “primarily in the form of dissolved...”

Response to reviewer comment No.3: Changed as you suggested.

(4) Please refer to phytoplankton as a singular noun throughout the

manuscript; otherwise refer to phytoplankton species or organisms

Response to reviewer comment No.4: Changed as you suggested.

(5) Line 66 = within phytoplankton, the species in which P regulation has been best studied is...

Response to reviewer comment No.5: Changed as you suggested.

(6) Line 68-69 = already stated in the introduction. Please remove “and it is a species of diatoms, a dominant group of phytoplankton contributing about 40% of marine primary production”

Response to reviewer comment No.6: Removed as you suggested.

(7) Line 70-71 = “has been identified by gene knockout experiments, with the PHR protein acting as a positive regulator of Pi signalling”

Response to reviewer comment No.7: Changed as you suggested.

(8) Line 86 = to examine the distribution and expression of the SPX. What do you mean here?

Response to reviewer comment No.8: From these three databases, we want to study the expression patterns, taxonomic affiliation, and geographical distribution of SPX domain-containing genes to find the evidence that this SPX-based P-homeostasis regulatory mechanism might exist in all major lineages of phytoplankton and is at play throughout the global oceans.

(9) Line 93 = please replace “Gianluca” with Dell’Aquila et al.

Response to reviewer comment No.9: Changed as you suggested.

(10) Line 94 = “verified the presence” please replace with “allowed the identification”

Response to reviewer comment No.10: Changed as you suggested.

(11) Line 95 = you can shorten the sentence stating that the sequences of *P. tricornutum* shared several conserved sites with land plants' SPX domains

Response to reviewer comment No.11: Changed as you suggested.

(12) Line 96 = it is not "affiliation", you may want to use phylogenetic closeness, similarity

Response to reviewer comment No.12: We have changed it to "high similarity".

(13) Line 256 = non-sense sentence. I believe there is a missing verb

Response to reviewer comment No.13: Added as you suggested.

**(14) Line 262 = the subject is mining, therefore the verb cannot be "indicate".
"Furthermore, mining of the TARA Oceans metatranscriptomic data, we clearly found evidence that this SPX-based P-homeostasis regulatory mechanism exists..."**

Response to reviewer comment No.14: Changed as you suggested.

(15) Line 264 = oceans

Response to reviewer comment No.15: Changed as you suggested.

(16) Line 387 = Guillard's F/2 medium. Add also the reference: Guillard, R. R. L. Culture of Phytoplankton for Feeding Marine Invertebrates. in Culture of Marine Invertebrate Animals: Proceedings — 1st Conference on Culture of Marine Invertebrate Animals Greenport (eds. Smith, W. L. & Chanley, M. H.) 29–60, https://doi.org/10.1007/978-1-4615-8714-9_3 (Springer US, 1975).

Response to reviewer comment No.16: Changed as you suggested.

(17) Line 392 = except that in P treatments, where phosphate concentrations provided in the media varied

Response to reviewer comment No.17: Changed as you suggested.

(18) Line 395 = “a P+ culture” – a P-replete culture?

Response to reviewer comment No.18: Changed as you suggested.

(19) Line 410 -412 = please put this sentence soon after the one above and add a new paragraph line after. If I have properly understood, at lines 405-412 you are summarising the protocol and from 413 onward you are providing details.

Response to reviewer comment No.19: Changed as you suggested.

(20) Line 410 -412 = At this point, after selecting putative homologs of SPX by genome blast, why the authors did not also blasted this sequence against MMETSP transcriptomes and Tara/MGT metatranscriptomes and infer a phylogenetic tree with all these sequences? This approach could have been of help in ascertaining the identity of the sequence.

Response to reviewer comment No.20: Here, our main goal was to determine the identity of the six SPX domain-containing genes in *P. tricornutum*. And blasting these sequences against NCBI database was to collect previously characterized SPX genes, which are necessary to build a tree. Subsequent blast against the MMETSP transcriptomes and Tara/MGT metatranscriptomes was to determine if SPX occurs widely geographically and taxonomically in general, which should be done after the function of SPX have been proven using the gene knockout work. Therefore, the latter part was done separately. To maintain an understandable logical flow, we need to keep these two parts separate.

(21) Line 412 = please specify at what similarity threshold (e-value) the search was performed.

Response to reviewer comment No.21: Added as you suggested.

(22) Line 407 = please specify the Pfam number used to allow reproducibility

Response to reviewer comment No.22: The full-length protein sequences were performed to Pfam search to examine whether the six genes identified contained only SPX domain or additional domains were present.

(23) Line 415 = please revise this sentence. The aLRT (approximate likelihood-ratio test) is a test to perform the support to branches. The default option for PhyML is the Shimodaira-Hasegawa (SH) like test. Did you perform this test? Please also make clear that you have inferred a maximum likelihood phylogenetic tree in PhyML, that was ran as daughter process of SeaView (correct the name in the text).

Response to reviewer comment No.23: Revised as you suggested. Prior to maximum likelihood (ML) phylogenetic analysis, the best amino acid substitution model fitting for SPX sequences was LG+G. ML phylogenetic trees were created using MEGA X with 1,000 bootstraps. Neighbor-Joining analysis conducted using MEGA X yielded similar tree topology.

(24) Line 418-419 = "expression dynamics of the identified SPX domain-containing genes were examined from transcriptomes sequenced under P+ and P- conditions". I do not understand the meaning of this sentence with a reference here. Do the authors mean that they have analysed the expression profiles of previously published transcriptomes? Why the reference? If so, they should argue better the reasons for this choice. If not, they should rephrase the sentence because it is not clear.

Response to reviewer comment No.24: Yes, we used our previous transcriptome data to investigate the P stress-induced SPX genes from the six identified SPX domain-containing genes. Based on the fact that SPX gene (Phatr3_J47434) can be induced by P stress and contains SPX domain as the sole functional domain, it was selected for mutagenesis. And in the new version, the relevant description has been added in the results section (page 5, lines 109-112), and we deleted the description in M&M (page 17, lines 471-475) to avoid confusion.

(25) Line 434 = at the beginning of M&M section there already is a paragraph about culture conditions. What experiments is it referring to? This is not clear. You may want to make a single paragraph on culture conditions for all the experiments instead of repeating it.

Response to reviewer comment No.25: Yes, we try to make a single paragraph on culture conditions for all the experiments at the beginning of M&M section. Here, *P. tricornutum* cells were collected from exponentially growing culture and were spread on agar plates for Gene Gun transformation. In the revised version, we simplified the description of this section to avoid confusing readers.

(26) Line 840 = conserved sites

Response to reviewer comment No.26: Changed as you suggested.

(27) Figure 4b = better to replace “phylogenetic maps....” with “Krona pie-charts showing the taxonomic distribution of SPX-domain containing genes”

Response to reviewer comment No.27: Changed as you suggested.

References

1. Love MI, Huber W, Anders S. Moderated estimation of fold change and dispersion for RNA-seq data with DESeq2. *Genome Biol* **15**, 550 (2014).
2. Nymark M, Sharma AK, Sparstad T, Bones AM, Winge P. A CRISPR/Cas9 system adapted for gene editing in marine algae. *Sci Rep* **6**, 24951 (2016).